# Development of an omnidirectional rotating Compton camera for imaging 177Lu radioactive contamination

Hikari Tsukamoto[1,6]*, Hiroshi Muraishi[1]*, Ryoji Enomoto[1], Hideaki Katagiri[2], Mika Kagaya[3], Takara Watanabe[1], Takahiro Mizoguchi[1], Masaya Fukumoto[1], Daisuke Kano[4], Yusuke Watanabe[1], Kazuya Sakaguchi[1], Hiromichi Ishiyama[5]

1 School of Allied Health Sciences, Kitasato University, Sagamihara, Kanagawa, Japan, 2 College of Science, Ibaraki University, Mito, Ibaraki, Japan, 3 National Institute of Technology, Sendai College, Sendai, Miyagi, Japan, 4 Department of Pharmacy, National Cancer Center Hospital East, Kashiwa, Chiba, Japan, 5 Department of Radiology and Radiation Oncology, Kitasato University, Sagamihara, Kanagawa, Japan, 6 Department of Radiological Technology, Tokai University Hospital, Isehara, Kanagawa, Japan

* tsukamoto.hikari@st.kitasato-u.ac.jp (HT); muraishi@ahs.kitasato-u.ac.jp (HM)

**Editor:** Hesham M.H. Zakaly, Ural Federal University named after the first President of Russia B N Yeltsin Institute of Physics and Technology: Ural'skij federal'nyj universitet imeni pervogo Prezidenta Rossii B N El'cina Fiziko-tehnologiceskij institut, RUSSIAN FEDERATION

## Abstract

In this study, we developed an omnidirectional rotating Compton camera that was capable of imaging low-level radioactive contamination caused by 177Lu-oxodotreotide, a novel radiopharmaceutical that has recently been attracting attention in nuclear medicine. The detector employs a compact design that comprises only six scintillator crystals mounted on a motorized rotating stage. By optimizing the crystal type and size, and optimizing the interval between crystals, the detector is able to adapt to a wide range of environmental conditions, including observable gamma-ray energies, dose rates, and angular resolution. Monte Carlo simulations using Geant4 were conducted to optimize the configuration of the detector. Based on the results of the simulation, a prototype detector using six 3.5 cm cubic CaF$_2$(Eu) crystals was developed for visualizing 177Lu-contaminated sites. The experimental results demonstrated that the detector could successfully visualize an unsealed 177Lu-oxodotreotide source with high sensitivity without being affected by gamma rays from 99mTc, which is also present in nuclear medicine facilities. The developed rotating Compton camera technology is anticipated to serve as a reliable environmental monitoring tool in nuclear medicine facilities. Through its ability to rapidly detect radioactive contamination, this detector has the potential to reduce the radiation exposure risks for both medical professionals and the general public.

## 1 Introduction

Targeted Radionuclide Therapy (TRT) is a treatment approach that involves the administration of radiopharmaceuticals, which then selectively accumulate in target tissues and deliver focused radiation to the target areas for effective therapy [1]. Recently, the concept of "theranostics", which integrates therapeutic and diagnostic applications, has garnered significant

**Data availability statement:** All relevant data are within the paper and its Supporting Information files.

**Funding:** Funding was provided by the Japan Society for the Promotion of Science KAKENHI Grant (Nos. 19H04492 to HM and 24K21128 to TW, https://www.jsps.go.jp/english/e-grants/) and Special Research Grants of the School of Allied Health Sciences, Kitasato University (No. 2022-1001 to HM). The funders had no role in study design, data collection and analysis, decision to publish, or preparation of the manuscript.

**Competing interests:** The authors have declared that no competing interests exist.

attention in the field of TRT [2,3]. Theranostics combines the use of diagnostic and therapeutic radiopharmaceuticals in a seamless process to visualize diseased tissues, administer treatment, and evaluate therapeutic outcomes. One of the radiopharmaceuticals used in this therapy (see Table 1) is Lutetium-177 ($^{177}$Lu), which has the unique characteristic of emitting both beta rays for therapeutic effects and gamma rays for diagnostic purposes. The beta rays cause direct damage to the tumor tissues; the gamma rays can be detected from outside the body and enable the monitoring of the distribution of the drug after treatment. One example of a TRT pharmaceutical that uses $^{177}$Lu is $^{177}$Lu-oxodotreotide [4], which was approved in Japan in 2021. This treatment targets somatostatin receptor-positive neuroendocrine tumors, and it has gained attention as a new treatment method alongside surgery and chemotherapy. In this therapy, 7.4 GBq of $^{177}$Lu-oxodotreotide is injected into the patient, with 4–6 treatments typically given at intervals of 3–8 weeks. The physical half-life of $^{177}$Lu is 6.7 days. Most of the administered drug is excreted through urine, although some remains in the body and can be found in the blood, urine, and feces. Therefore, it should be noted that gamma rays and/or beta rays from $^{177}$Lu can also be emitted from the patient's body or from excreted contaminants, and it poses a risk of radiation exposure. Medical professionals who are involved in such treatment face occupational exposure risks, and the International Commission on Radiological Protection (the ICRP) recommends limiting occupational exposure to a maximum of 50 mSv per year and an average of 20 mSv per year over five years (i.e., 100 mSv over five years) [6]. This therapy requires either a Radioisotope (RI) treatment ward or a "special measures patient room" as specified in Article 30, Section 12 of the Ordinance for Enforcement of the Medical Services Act: Rooms for patients undergoing radiation therapy in Japan. RI treatment wards are equipped with radiation shielding, special exhaust, and drainage systems to manage radioactive contamination. Moreover, "special measures patient rooms" are general wards that are temporarily designated to serve as controlled areas. These rooms have radiation protection measures but are not equipped with the same systems as RI treatment wards. If adequate decontamination is not carried out after discharge, radioactive contamination may remain and present a risk of radiation exposure to subsequent patients and medical professionals. Furthermore, the cleaning of patient rooms is usually done by general cleaning staff rather than medical professionals, which means they are also subject to a risk of exposure. As the ICRP recommends limiting public exposure to 1 mSv per year [7], radiation protection measures are essential. However, RI treatment wards are extremely limited in Japan, and most medical facilities are forced to use special measures patient rooms. Therefore, for these wards to be used as general wards, they must be thoroughly decontaminated so that no radioactive

**Table 1. Radiopharmaceuticals used for TRT [1–3].**

| Radionuclide | $^{131}$I | $^{90}$Y | $^{89}$Sr | $^{177}$Lu | $^{223}$Ra |
|---|---|---|---|---|---|
| **Types of decay** | $\beta^-, \gamma$ | $\beta^-$ | $\beta^-$ | $\beta^-, \gamma$ | $\alpha, \gamma$ |
| **Half-life [day]** | 8.0 | 2.7 | 50.5 | 6.6 | 11.4 |
| **Primary gamma-ray energies [keV]** | 364(81.7%) | — | — | 208(11%) | 269(13.9%) |
| | | | | 113(6.4%) | 154(5.6%) |
| | | | | | 271($^{219}$Ra)* |
| | | | | | 405($^{211}$Pb)* |
| | | | | | 351($^{211}$Bi)* |
| **Primary beta(alpha)-ray energies [MeV]** | 0.606(89.5%) | 2.280(100%) | 1.495(100%) | 0.498(78.6%) | 5.7(51.6%) |
| | | | | 0.176(12.2%) | 5.6(25.2%) |
| **Max.(avg.) range in tissue [mm]** | 2(0.6) | 11(5.3) | 8(2.4) | 1.7(0.23) | < 0.1 |

**Note:** *These values represent the main gamma-ray energies emitted during the decay of $^{223}$Ra as it undergoes alpha and beta decay to ultimately become $^{207}$Pb [5].

materials remain. Given these requirements, strict radiation protection and decontamination measures must be established for the safe implementation and widespread use of $^{177}$Lu therapy.

Currently, the primary means of detecting radioactive contamination in medical settings are area monitors and survey meters. However, given that the anticipated radioactive contamination in nuclear medicine facilities is comprised of low doses (around 10 $\mu$Sv/h or less), area monitors that are designed for higher doses are unsuitable. Therefore, the primary method for detecting radioactive contamination is the survey meter, but if the radioactive contamination is not easily identified, a manual search of each location is needed, which is time-consuming and labor-intensive. Moreover, considering that footwear is the main source of radioactive contamination, this method has its limitations. The problem of radioactive contamination detection can be resolved by introducing a gamma-ray visualization device that is able to monitor large areas over a short time with high sensitivity. Such a device could serve as an effective alternative to area monitors and survey meters. Additionally, if the device is made portable, it becomes easier to use and poses a minimal burden when it is implemented in various facilities. The rapid identification of contaminated areas would improve work efficiency and, particularly in nuclear medicine facilities, help reduce the risk of radiation exposure. A high-sensitivity gamma-ray visualization device could play an important role in strengthening radiation protection and ensuring the safety of medical professionals and patients.

Gamma-ray detectors used for detecting radioactive contamination include pinhole-type cameras [8], coded-aperture gamma-ray cameras [9–11], and Compton cameras [12–14]. Pinhole-type and coded-aperture cameras have low sensitivity due to their lead-shielded structure, and their field of view is limited. Additionally, they tend to be bulky, which makes them impractical and difficult to move around. However, Compton cameras do not require any physical shielding such as lead shielding, and they more easily achieve a wide field of view and are low in weight. They are also capable of high-sensitivity imaging for low-dose radioactive contamination. This technology has been developed in the field of gamma-ray astrophysics [15–19] and is now widely used in medical diagnostics [20–22] and environmental radiation imaging [23–31]. As an example of a gamma-ray detector for environmental radiation imaging, Polaris-H was developed by the University of Michigan and commercialized by H3D [23]. It uses CdZnTe (CZT) and offers excellent energy resolution ($\Delta$E/662 keV = 1.1% FWHM). This system can achieve relatively high sensitivity as a Compton camera at energies above 250 keV, making it particularly useful for identifying radioactive contamination in nuclear power plants. However, at energies below 250 keV, it operates as a coded-aperture gamma-ray camera, which limits its field of view and sensitivity.

Recently, we proposed and developed a novel omnidirectional rotating Compton camera technology for environmental radiation monitoring [32]. Fig 1 shows a conceptual diagram of the technology. A Compton camera typically employs a detector with a two-layer structure that consists of a scatterer and an absorber to estimate the direction of incoming gamma rays using Compton kinematics. This technique employed the twofold coincidence within six CsI(Tl) scintillator cubes of 3.5 cm size, so that all crystals acted as both scatterers and absorbers, thereby enabling omnidirectional imaging. An incident gamma ray undergoes Compton scattering within the first crystal and is absorbed by the second crystal via the photoelectric effect. By measuring the energy deposited during these two interactions, the scattering angle $\theta_{\text{scat}}$ is calculated using the following equation:

$$\theta_{\text{scat}} = \arccos\left[1 - m_e c^2 \left(\frac{1}{E_2} - \frac{1}{E_1 + E_2}\right)\right]. \tag{1}$$

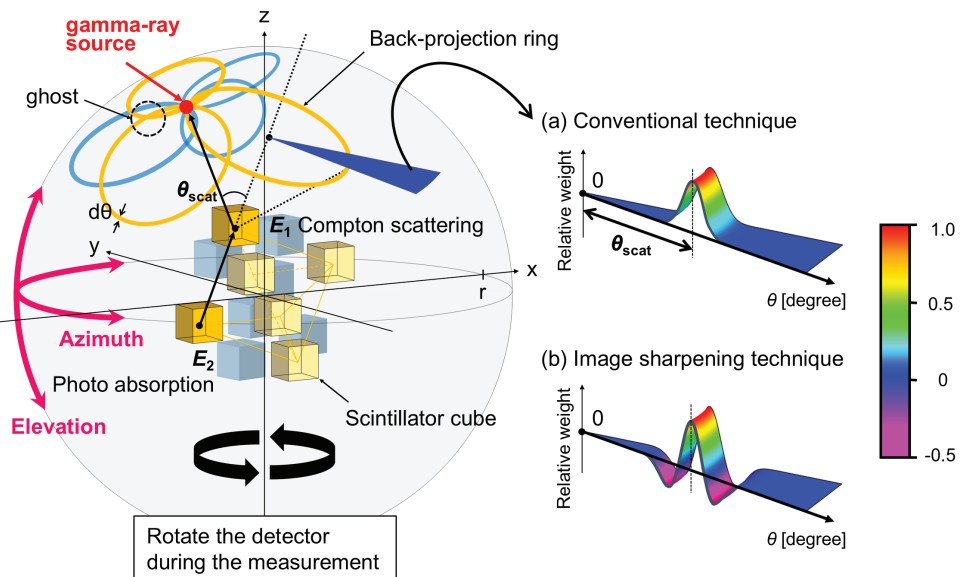

**Fig 1. Conceptual diagram of the omnidirectional rotating Compton camera technology** [32]**.** Six scintillators were placed at the vertices of an octahedron. By rotating the detector during the measurement, the number of crystals can be virtually increased, effectively eliminating the "ghosts" that occur when there are a small number of back-projection ring patterns in the reconstructed image. A smeared ring with a radius of scattering angle $\theta_{scat}$, which was calculated from the Compton kinematics for twofold coincidence events, was back-projected onto a spherical surface having a radius of $r$. The right figure represents the cross-section of a smeared ring, (a) the conventional technique (a Gaussian profile), and (b) the image sharpening technique (a multiple-Gaussian profile).

Here, $E_1$ and $E_2$ are the energies deposited by the recoil electron in the scatterer and the scattered gamma ray in the absorber, and $m_e c^2$ represents the rest mass energy of an electron. The gamma-ray image was then reconstructed by back projection as a ring shape with a radius of $\theta_{scat}$ for each event onto a spherical surface with radius $r$. However, using a small number of crystals results in a small number of ring patterns, which causes artificial uneven structures (called "ghosts") in the reconstructed image. To overcome this drawback, the number of crystals can be virtually increased by rotating the detector during the measurement, which successfully removes "ghosts" from the entire field of view. Furthermore, two reconstruction techniques have been used with this technology: the conventional technique and the image sharpening technique. With the conventional technique, a smeared ring based on a Gaussian profile, as shown in Fig 1a, is used to overlay positive weights in the reconstructed region. Although this method produces a reconstructed image that has poor angular resolution, it is excellent for identifying the direction of the gamma-ray peak intensity within a short time. However, the image sharpening technique, which is based on the filtered back-projection algorithm used in computed tomography, overlays positive and negative weights on the reconstructed region using a smeared ring as shown in Fig 1b. As a result, it is possible to suppress the background region to the gamma-ray source to a zero level while sharpening the gamma-ray peak. Another notable feature of the Compton camera technology described above is that the detector can be customized to the appropriate conditions (gamma-ray energy, dose rate, angular resolution, etc.) by optimizing the crystal size and type, and the interval of the crystals, and it can be developed at low cost.

In this study, we explored the application of the omnidirectional rotating Compton camera technology [32] to the visualization of the radioactive contamination caused by $^{177}$Lu. In Sect 2, we first described the Monte Carlo simulations that were carried out to optimize the detector configuration. Based on these findings, we present in Sect 3 a prototype detector and assess its performance through $^{177}$Lu imaging experiments at a nuclear medicine facility. Finally, the broader implications and potential applications of this technology are discussed in Sect 4.

## 2 Detector design

### 2.1 Simulation overview

In this study, we first investigate the applicability of the omnidirectional rotating Compton camera technology [32] by performing Monte Carlo simulations. Fig 2 shows a schematic of the crystal arrangement used in this simulation. The six crystals were placed at the vertexes of an octahedron having a side length of $W$ cm. The crystal cube size was defined as $3.5 \times (W/10)$ cm. For example, if we assume $W = 10$ cm, the crystal size is 3.5 cm, which corresponds to the same condition as in the previous study [32]. Because the crystal cube size is scaled according to the interval of the crystals, the angular resolution of the reconstructed image is expected to be $\sigma \sim 10$ degrees, due to the fact that the geometrical factor that results from the solid angle between the crystals is dominant, as in the previous study [32]. When the angular resolution of $\sigma \sim 10$ degrees is achieved, a gamma-ray source that is 1.5 m away can be identified with an accuracy of 30 cm, which is considered appropriate for the purpose of this study.

Using the framework described above, simulations were conducted for various combinations of crystal types, sizes, and gamma-ray energies. The selected crystal types included CsI, NaI, and CaF$_2$, and their respective characteristics are summarized in Table 2. These crystals were selected because of their high-energy resolution and excellent cost performance. The crystal sizes were set to between 0.5 and 4.0 cm, with a 0.5 cm interval between values. The gamma-ray energies investigated were 113 keV ($^{177}$Lu with a gamma-ray emission ratio $\eta_\gamma = 0.064$), 208 keV ($^{177}$Lu with a gamma-ray emission ratio $\eta_\gamma = 0.11$), 511 keV ($^{22}$Na with

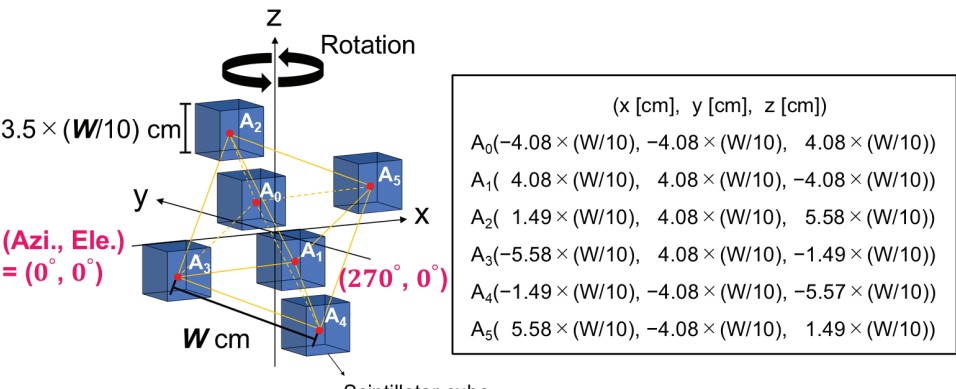

**Fig 2. Schematic view of the defined crystal arrangement in this simulation.** Six crystals were placed at the vertexes of an octahedron as in the previous study [32], where the crystal size was scaled to the side length $W$ cm of the octahedron.

**Table 2. Basic properties of scintillator crystals [33].**

|  | CsI(Tl) | NaI(Tl) | CaF$_2$(Eu) |
|---|---|---|---|
| Atomic number (Z) | 55, 53 | 11, 53 | 20, 9 |
| Effective atomic number (Z$_{eff}$) | 54 | 47 | 15 |
| Density [g/cm$^3$] | 4.51 | 3.67 | 3.19 |
| Decay time [μs] | 0.68(64%) | 0.23 | 0.9 |
|  | 3.34(36%) |  |  |
| Wavelength of Max. emission [nm] | 540 | 415 | 435 |
| Abs. light yield [photons/MeV] | 65000 | 38000 | 24000 |

a gamma-ray emission ratio $\eta_\gamma$ = 2.00), and 1333 keV ($^{60}$Co with a gamma-ray emission ratio $\eta_\gamma$ = 1.00). The simulations were conducted in a virtual environment built with VMware using Geant4 (version 10.7 patch01) [34]. The host OS was Windows 10 (Intel Core i5 with a 1.6 GHz processor), and the virtual environment used the Ubuntu (version 20.04) OS. Additionally, FTFP-BERT [35] was used for physics lists in Geant4.

For the simulation, the initial position of the gamma rays with energy $E_\gamma$ was assumed to be $I$(−100 cm, 0 cm, 0 cm) to evaluate the reconstructed image of a point source in the direction of (azimuth, elevation) = (0 degrees, 0 degrees). Here, to simulate the rotating Compton camera, the position of the crystals had to be rearranged for each gamma-ray event by rotating it on the horizontal plane. However, Geant4 did not allow the position of the crystal to be redefined for each event. Therefore, instead of fixing the coordinates of the crystal, we redefined the starting position $I$ by rotating it around the detector for each event, which allowed us to simulate a rotating Compton camera: $I$ is redefined by rotating it on the horizontal plane using the angle $\theta_{rot}$ obtained uniformly between 0 degrees and 360 degrees, and then the gamma ray is emitted isotropically. Subsequently, if the gamma ray interacted with two of the six crystals, the energy $E$ deposited at each crystal and $\theta_{rot}$ were saved for each event. Finally, to reproduce the actual measurements, $E$ was added with a random number following a Gaussian distribution with a standard deviation $\sigma(E)$. Here, we used $\sigma(E) = 0.5746E^{0.5736}$, which is the energy resolution obtained from the measurements using a cubic CsI(Tl) crystal with sides of 3.5 cm coupled with a photomultiplier tube (PMT) [27] and is assumed to be a typical energy resolution for the scintillation detector used in this study.

## 2.2 Image reconstruction

Next, we select the events to be used for image reconstruction from the twofold coincidence events obtained from the Monte Carlo simulation described in the previous section. Gamma-ray events are selected using two criteria: time lag selection (TDC Cut) and energy selection (ADC Cut). For the TDC Cut, events having a time lag within ±400 ns are identified as twofold coincidence events, whereas accidental coincidences are excluded, following the analysis of real data from a previous study [32]. This step can be omitted for data derived from the simulation because each event is simulated individually. For the ADC Cut, the reconstruction energy is selected from the total energy spectrum of the twofold coincidence events (the sum of $E_1 + E_2$). Fig 3 shows the distribution of the total energy of the 208 keV gamma rays obtained from the simulation as an example. The total absorption peak of the 208 keV gamma rays can be seen clearly in this figure. Gamma-ray events within ±3$\sigma$, as indicated by the red shaded area in Fig 3, were selected as the events to be used for the image reconstruction, in which the smeared rings were back-projected onto the surface of a sphere having a radius of

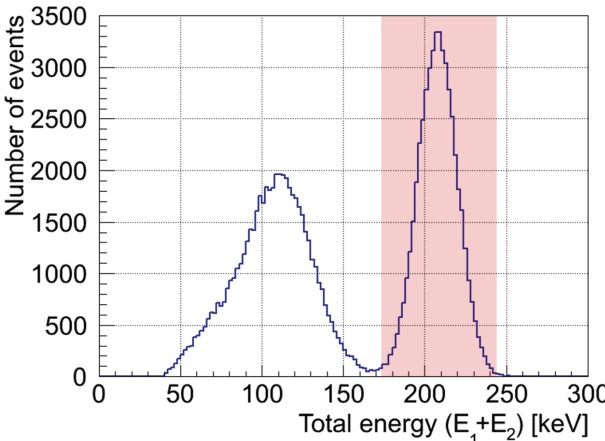

**Fig 3. Example of the total energy spectrum (the sum of $E_1 + E_2$) of 208 keV gamma rays derived from the simulation assuming a Compton camera with six 3.5 cm cubic CaF$_2$ crystals.** The red shaded area represents the energy window for selecting events used for the image reconstruction of 208 keV gamma rays, which is set to $208 \pm 3\sigma$ keV.

1 m. To reproduce the rotating Compton camera in this simulation, the starting position of the gamma ray $I$ was rotated randomly by $\theta_{\text{rot}}$ for each event, as was described in the previous section. Therefore, for the image reconstruction, the crystal coordinates were rotated by $-\theta_{\text{rot}}$ on the horizontal plane for each event, and then the smeared ring was back-projected onto a spherical surface.

As shown in Fig 1, the shape of the smeared ring should differ between the conventional and image sharpening techniques [32]. In this study, we defined the 1D profiles of the smeared ring for the conventional technique $G_0(\theta)$ and the image sharpening technique $G_1(\theta)$ as a function of $\theta$ as follows:

$$G_0(\theta) = \exp\left(-\frac{(\theta - \theta_{\text{scat}})^2}{2\sigma^2}\right), \tag{2}$$

$$
\begin{aligned}
G_1(\theta) = k \times \Bigg\{ & 2\exp\left(-\frac{(\theta - \theta_{\text{scat}})^2}{2\sigma^2}\right) \\
& - \exp\left(-\frac{(\theta - \theta_{\text{scat}} - 2\sigma)^2}{2\sigma^2}\right) \\
& - \exp\left(-\frac{(\theta - \theta_{\text{scat}} + 2\sigma)^2}{2\sigma^2}\right) \Bigg\}.
\end{aligned}
\tag{3}
$$

Here, $\theta_{\text{scat}}$ represents the scattering angle calculated from Eq (1) with the energy deposits $E_1$ and $E_2$ in the two crystals during the event selected after the ADC cut described above. In this study, we assume $\sigma = 8$ degrees based on previous studies [27,32]. Additionally, $k$ is a scaling factor expressed as $k = N/V(N)$, where $N$ is the number of events used for image reconstruction, and $V(N)$ is the peak value when the smeared ring with $k = 1$ in Eq (3) is reconstructed using $N$ events. By deriving $k$ obtained with $N = 50000$ events in advance, the image reconstructed using the image sharpening technique can be expressed as a distribution of the absolute values of the detected gamma-ray intensity.

## 2.3 Evaluation indices

To investigate how the characteristics of the Compton camera are affected under each condition in this simulation, a quantitative evaluation was performed for the angular resolution, contrast, and sensitivity. In this study, the following three evaluation indices were adopted.

First, the angular resolution was defined as the standard deviation $\sigma$ obtained by fitting the reconstructed image with a 2D Gaussian function $f(\theta_{azi}, \theta_{ele})$ as

$$f(\theta_{azi}, \theta_{ele}) = p \exp\left(-\frac{\theta_{azi}^2 + \theta_{ele}^2}{2\sigma^2}\right), \tag{4}$$

where $\theta_{azi}$ and $\theta_{ele}$ represent the azimuth and elevation angles of the reconstructed image in degrees. Here, $p$ and $\sigma$ are the free parameters obtained from the fitting.

Second, the contrast was evaluated using the Contrast-to-Noise Ratio (CNR), which was calculated as follows:

$$CNR = \frac{Contrast}{Noise} = \frac{P - M_{BG}}{\sigma_{BG}}, \tag{5}$$

where $P$ represents the maximum pixel value in the reconstructed image, and $M_{BG}$ and $\sigma_{BG}$ are the mean and standard deviation of the pixel values in regions that exclude the region centered on the source direction and having a 30-degree radius.

Third, the detector's sensitivity was quantified using the absolute sensitivity at a source-detector distance of 1 m and was defined as:

$$Absolute\ sensitivity = \frac{C(r)}{C_0}, \tag{6}$$

where $C_0$ represents the total number of gamma-ray events isotropically emitted from a point source, and $C(r)$ represents the number of events utilized for image reconstruction after ADC Cut as a function of the distance $r$ between the gamma-ray source and the detector; $r$ is set to 1 m for this simulation.

## 2.4 Simulation results

Fig 4 shows the reconstructed images for gamma rays having energies of (a) 113 keV ($^{177}$Lu), (b) 208 keV ($^{177}$Lu), (c) 511 keV ($^{22}$Na), and (d) 1333 keV ($^{60}$Co), which were obtained by the simulation. Here, the image reconstruction was performed using the image sharpening technique. The images are displayed with azimuth $\theta_{azi}$ on the horizontal axis and elevation $\theta_{ele}$ on the vertical axis in degrees. The matrix size is 180 × 90, which corresponds to 2 degrees for each bin. The top row presents the results for a 1.5 cm cubic crystal, and the bottom row shows the results for a 3.5 cm cubic crystal. Each column represents, from left to right, the results for CsI, NaI, and CaF$_2$. We observe that the omnidirectional rotating Compton camera technique investigated in this simulation is capable of visualizing gamma rays over a wide energy range. Notably, the 3.5 cm cubic CsI results for 511 keV gamma rays (Fig 5c) demonstrates performance comparable to that of a detector developed in a previous study [32]. Furthermore, the angular resolution and contrast are comparable at each energy, regardless of the crystal type or size.

The angular resolution $\sigma$ and CNR were estimated by applying Eqs (4) and (5) to the results shown in Fig 4. The results are shown, respectively, in the left and middle columns of

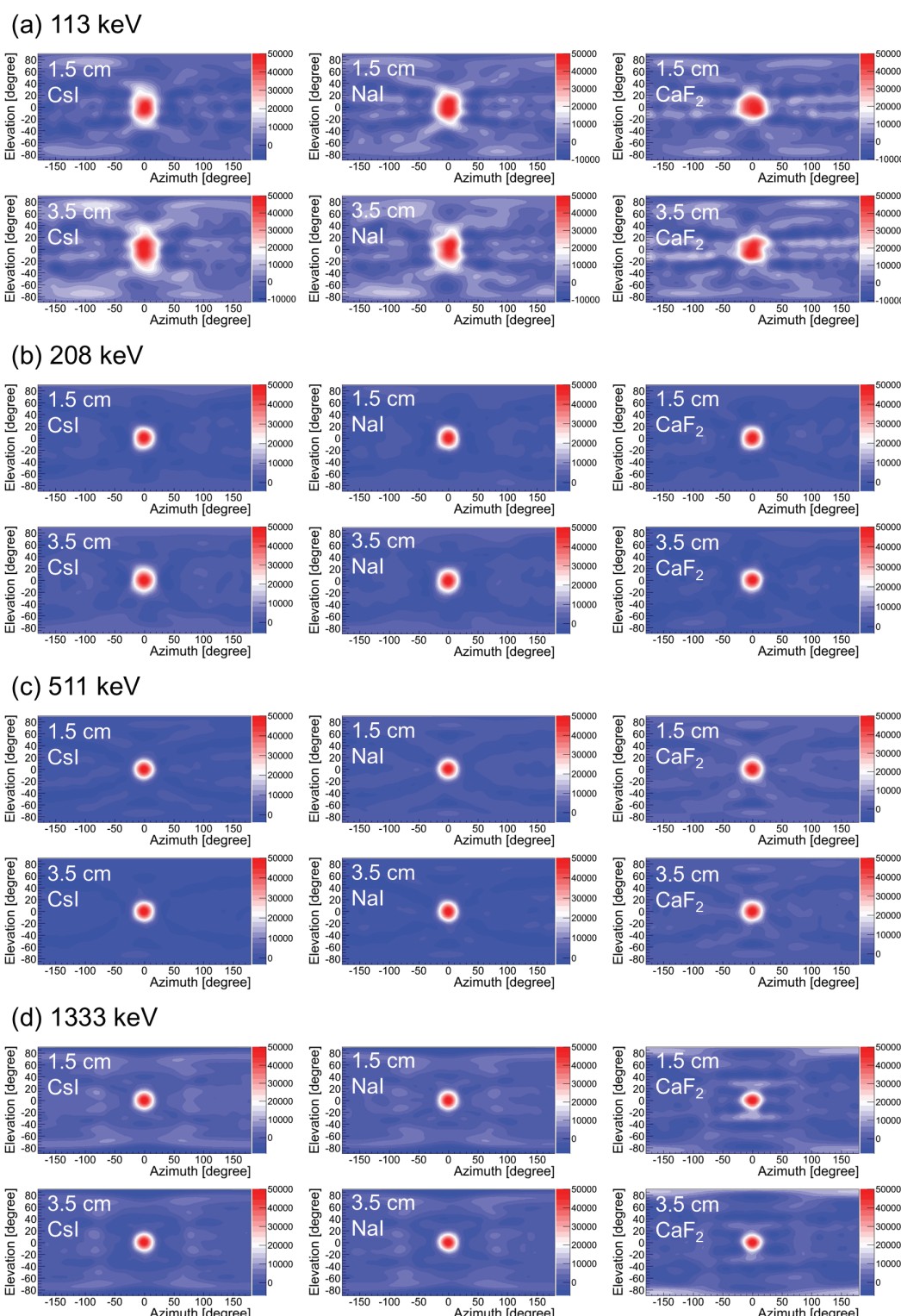

**Fig 4. Expected omnidirectional reconstructed images from a hypothetical gamma-ray point source in the direction of (azimuth, elevation) = (0 degrees, 0 degrees) obtained by image sharpening technique (distance between detector and source: 1 m).** (a) 113 keV, (b) 208 keV, (c) 511 keV, (d) 1333 keV. The left, middle, and right columns represent the results for CsI, NaI, and CaF$_2$, respectively. The upper and lower rows represent the results for the 1.5 cm and 3.5 cm cubic crystals, respectively. The number of events used for the back-projection was fixed at 50,000 events for all cases.

Fig 5 as a function of crystal size. The angular resolution $\sigma$ was fairly constant for all crystal types and sizes, and it was approximately 15 degrees at 113 keV and 11 degrees elsewhere. This is consistent with our expectation that the angular resolution $\sigma$ could be made constant by scaling the crystal size with the crystal intervals. The CNR also tended to be roughly constant, similar to the angular resolution $\sigma$. In particular, the CNR of $CaF_2$ was high at 208 keV. These findings suggest that the influence of the crystal type and size on the image quality is minimal at each energy level.

The absolute sensitivity was estimated for the data shown in Fig 4 by using Eq (6). The results are shown in the right column of Fig 5. The absolute sensitivity is expected to increase with the crystal size over a wide energy range. In particular, $CaF_2$ showed higher sensitivity than did NaI or CsI for crystal sizes larger than 1 cm at 113 keV and crystal sizes larger than 2.5 cm at 208 keV. To demonstrate how the absolute sensitivity changes with energy level, we redisplayed the absolute sensitivity shown in the right column of Fig 5 as a function of energy. The results are shown in Fig 6. We can see that CsI and NaI exhibit higher sensitivity in the high-energy region above 300 keV, whereas $CaF_2$ shows excellent sensitivity in the low-energy region below 300 keV.

Fig 7 shows the elevation dependence of the absolute sensitivity obtained from simulations with different elevation angles of the assumed gamma-ray source. Here, the values are normalized to those for elevation = 0 degrees for each condition. We remark that because the crystal arrangement is symmetric about the origin, the results are the same whether the elevation is positive or negative. It can be seen that the variation is 30% at 113 keV, 15% at 141 keV, and generally uniform within about 5% at 208 keV and above.

Based on these results, we conclude that the detector containing six 3.5 cm cubic $CaF_2$ crystals is the most suitable choice for visualizing low-level radioactive contamination from $^{177}$Lu, when portability and cost performance are also considered. Notably, the simulation results shown above were consistent even when detailed simulations were performed using a physics list, such as LIVEMORE [35], which was specifically designed for low-energy gamma rays. Furthermore, as described in Sect 2.1, the energy resolution $\sigma(E)$ obtained by measuring CsI(Tl) was employed in simulations using NaI(Tl) and $CaF_2$(Eu). Nonetheless, the simulation results were consistent even when the energy resolution obtained by measuring 3.5 cm cubic $CaF_2$(Eu) coupled with a metal-packaged PMT ($\sigma(E) = 1.001E^{0.4963}$, which is 10% larger than that for CsI(Tl)), was used. This implies that the angular resolution of the Compton camera used in this simulation is dominated by the geometrical factor, consistent with a previous study [32].

## 3 Prototype detector

### 3.1 Overview

Based on the simulation results presented earlier, we developed a prototype detector employing six 3.5 cm cubic $CaF_2$(Eu) crystals. Fig 8 shows photographs of the detector and the counter developed in the present study. The counter consists of a $CaF_2$(Eu) crystal (OKEN, cube size: 35 mm × 35 mm × 35 mm) coupled with a metal-packaged PMT (Hamamatsu Photonics, H13543-100, size: 33 mm × 33 mm × 50 mm). According to the design of the simulation, six crystals were placed at the vertices of an octahedron having a side length of 10 cm, which was then mounted on a motorized rotation stage (Sigma Koki, OSMS-60YAW). To make the detector more compact, the PMTs used in previous studies (Hamamatsu Photonics H11432-100, $\varphi$39 mm × 128 mm) [32] were replaced with metal-packaged PMTs that were approximately half the size. The signals from the PMTs were first processed by an amplifier board for waveform shaping and amplification. The signals were then sent to a 16-channel

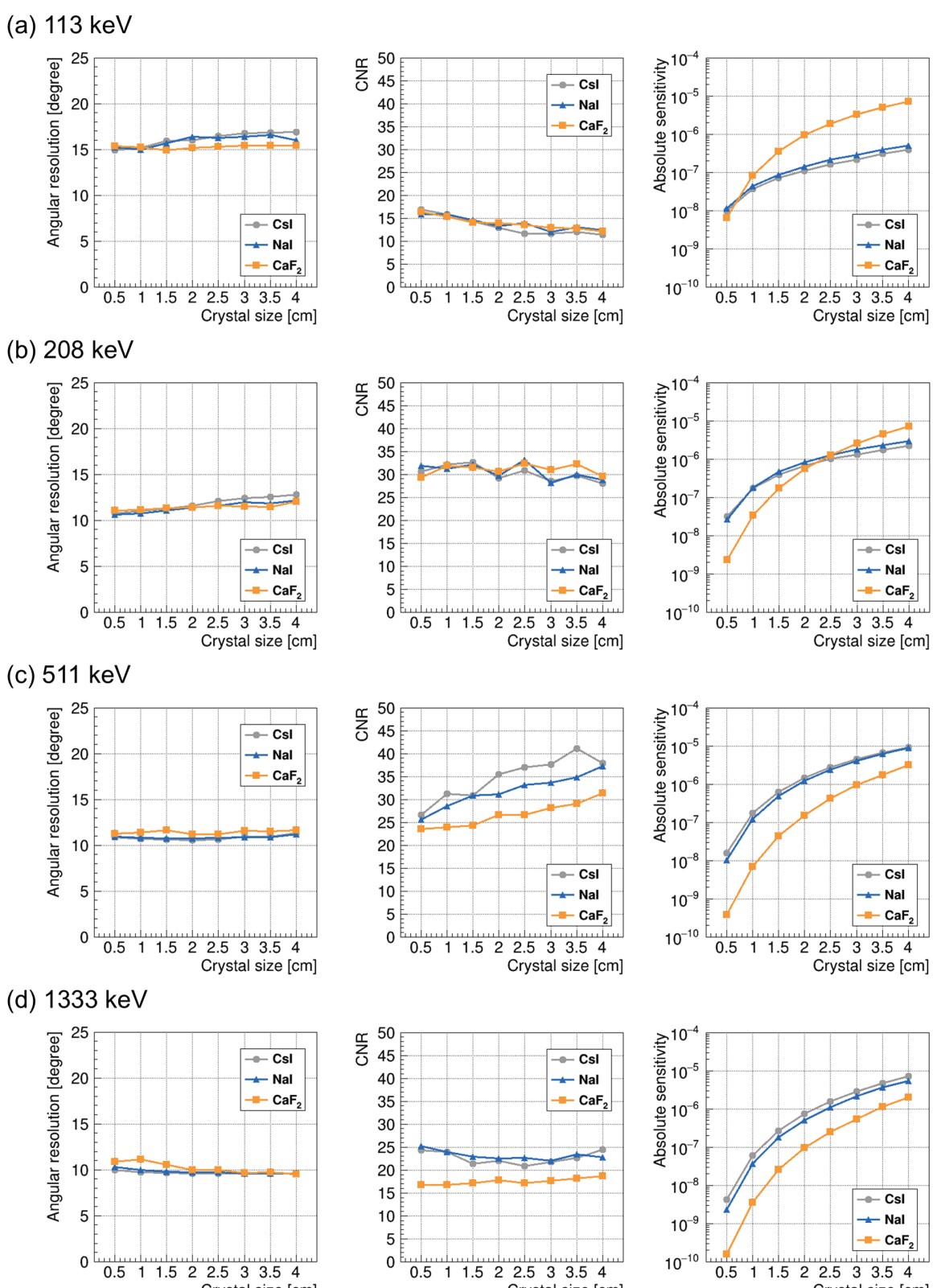

**Fig 5. Angular resolution $\sigma$ (left), CNR (middle), and absolute sensitivity (right) as a function of crystal size.** (a) 113 keV, (b) 208 keV, (c) 511 keV, (d) 1333 keV. The gray circle, blue triangle, and orange square represent the results for CsI, NaI, and CaF$_2$, respectively.

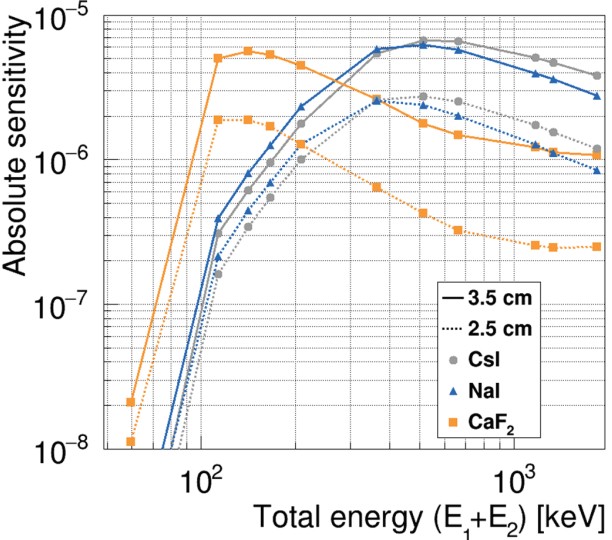

**Fig 6. Absolute sensitivity as a function of gamma-ray energy.** The gray circle, blue triangle, and orange square represent the results for CsI, NaI, and CaF$_2$, respectively. The solid and dotted lines indicate the case for crystal sizes of 3.5 cm and 2.5 cm cubed, respectively.

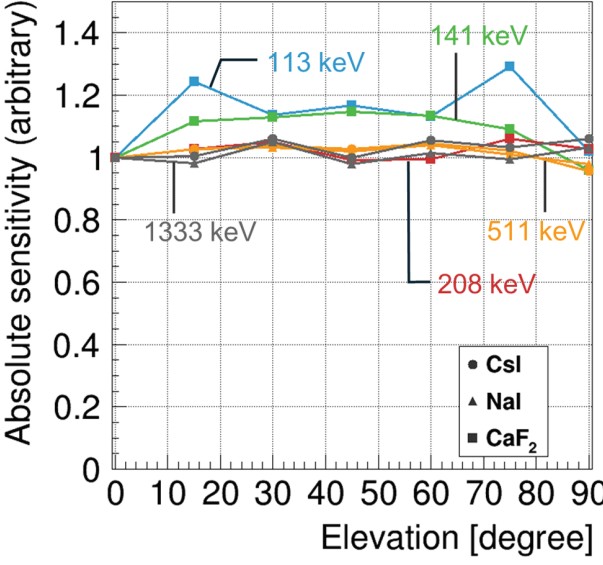

**Fig 7. Absolute sensitivity (arbitrary) as a function of the elevation angle of the assumed gamma-ray source.** The values were obtained using the same procedure as in Figs 5 and 6, but the position of the gamma-ray source differed in the elevation direction. The values were normalized to that at an elevation = 0 degrees for each condition. The circles, triangles, and squares represent the results for CsI, NaI, and CaF$_2$, respectively. The results for CaF$_2$ at 113 keV (blue), 141 keV (green), and 208 keV (red) are presented, as well as the results for CsI and NaI at 511 keV (orange) and 1333 keV (gray).

flash ADC board (operating at 2.5 MHz) using SiTCP technology [36]; when two hits coincidently occur above some threshold, a gate signal is generated, and the corresponding waveform data are stored in FPGA registers [26]. The stored data are then transferred to a PC via

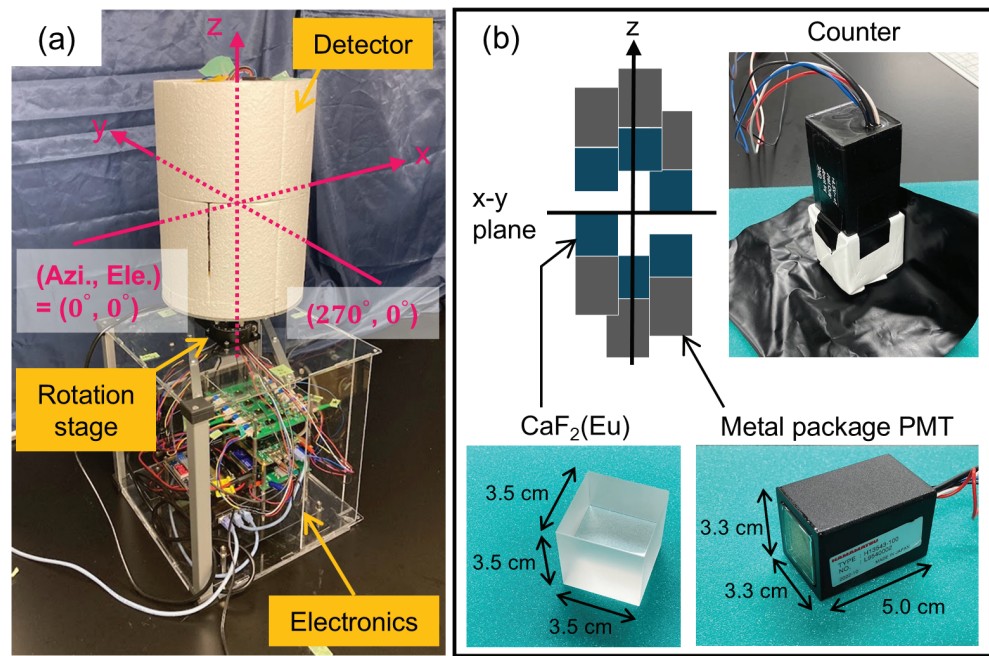

**Fig 8. Photograph of the omnidirectional rotating Compton camera developed in this study.** (a) Overall photo of the prototype detector. (b) Photo and layout of a counter consisting of a 3.5 cm cubic $CaF_2(Eu)$ coupled with a PMT.

an Ethernet connection; the online program, which is developed using Visual C++ and the ROOT library [37], is run on a Windows PC. The rotation stage is also controlled by an independent program written in Visual C++ via the Ethernet using a serial-to-Ethernet converter that converts RS232C to Ethernet, where the rotation angle is updated in a shared memory every 0.1 s. The online program saves not only the stored waveforms but also the present rotation angle for each event by referring to the shared memory. The equipment, except for the counters and the rotation stage, was housed in the acrylic case shown in Fig. 8a, which made it portable. As a result, the prototype detector had a size of $30 \times 30 \times 60$ cm and a weight of 7.2 kg, not including the lithium battery (approximately 1.7 kg).

## 3.2 Performance tests

The performance of the developed detector was evaluated by measuring a $^{177}$Lu unsealed source. First, we used $^{177}$Lu-oxodotreotide left in a glass vial after TRT (Fig 9a). The radioactivity was 43 MBq (CAPINTEC, CRC-55tR-type RI dose calibrator). Fig 9b shows the experimental setup, where the source was placed 1 m away from the detector at (azimuth, elevation) = (0 degrees, 0 degrees). The air dose rate during the measurement at the detector position was 0.19 $\mu$Sv/h, which included a background of 0.03 $\mu$Sv/h (Hitachi Aloka Medical, Ltd., TCS-172B). The measurement was conducted using continuous detector rotation, assuming the rotation stage moved back and forth between 0 degrees and 350 degrees at a constant speed and completed one cycle every 2 min. Next, the measurement of $^{177}$Lu was conducted under the assumption that the actual contaminated environment was in a nuclear medicine facility, specifically in the presence of $^{99m}$Tc (141 keV gamma rays), which is a commonly used diagnostic radiopharmaceutical. The sources used were $^{177}$Lu-oxodotreotide (radioactivity: 37.3 MBq) in a glass vial and $^{99m}$Tc (radioactivity: 5.3 MBq) in a medical syringe. The

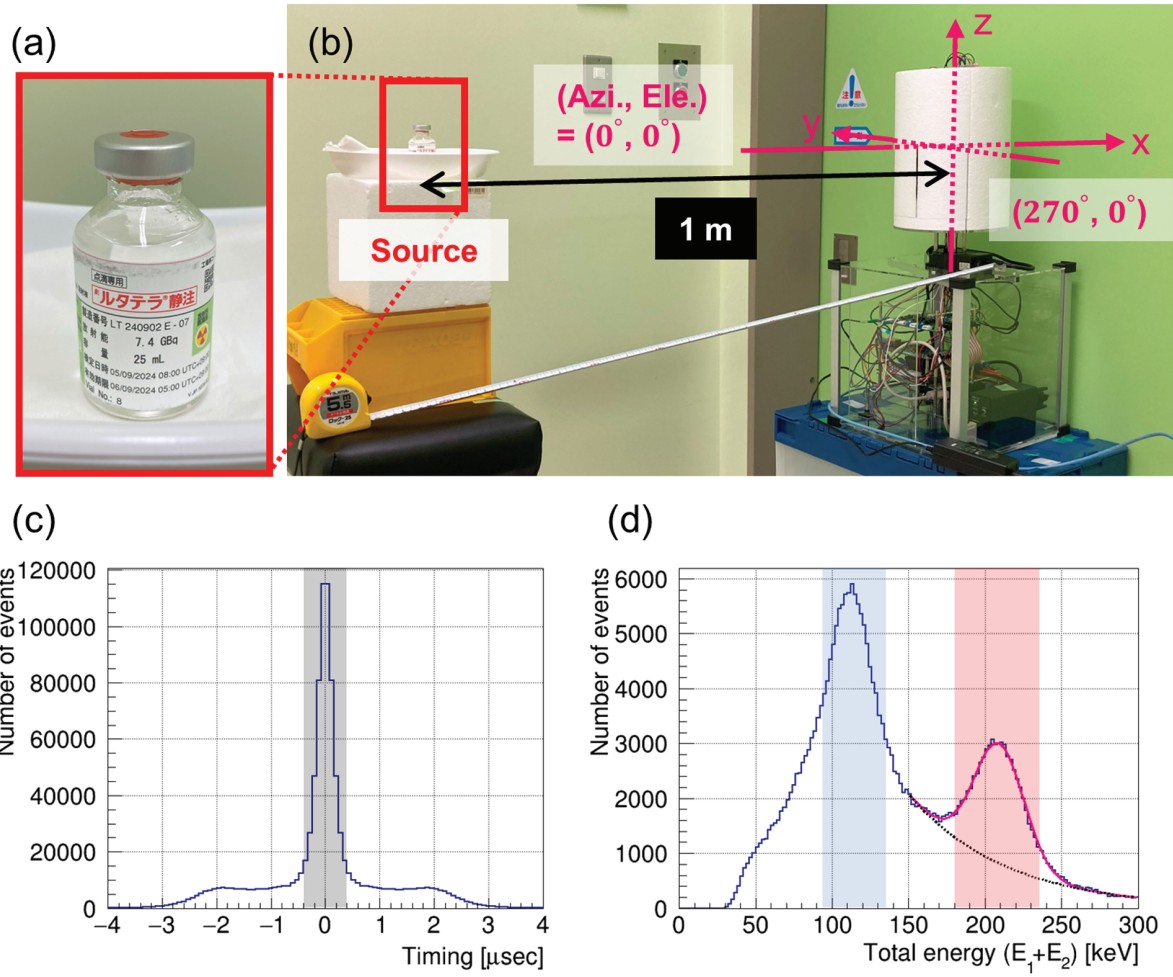

**Fig 9. Performance test with the $^{177}$Lu-unsealed source.** The source was placed 1 m ahead of the detector in the direction of (azimuth, elevation) = (0 degrees, 0 degrees). (a) Spent, unsealed $^{177}$Lu-oxodotreotide in a vial (43 MBq). (b) Photograph of the measurement setup. (c) Time lag distribution of twofold coincidence events. The gray shaded area represents the on-timing threshold for the TDC Cut with ±400 ns. (d) Total energy spectrum after the TDC Cut (the sum of $E_1 + E_2$). The spectrum shows peaks for 113 and 208 keV gamma rays. The blue and red shaded areas represent the ADC Cut with ±2σ energy regions, which were set to select events to be used for the image reconstruction of 113 keV and 208 keV gamma rays, respectively.

$^{177}$Lu source was placed 125 cm from the detector at (azimuth, elevation) = (50 degrees, −10 degrees), and the $^{99m}$Tc source was placed 150 cm from the detector at (azimuth, elevation) = (−60 degrees, −30 degrees). During the measurement, the air dose rate at the detector position was 0.21 $\mu$Sv/h. The continuous detector rotation setting was the same as previously described.

To prevent radioactive contamination in case of vial or syringe damage, the $^{177}$Lu and $^{99m}$Tc sources were each placed on trays lined with a filter paper during experiments. The operators wore gloves and used the personal dosimeters to ensure proper radiation protection. During measurements, the operators evacuated the controlled radiation area to minimize exposure as much as possible.

## 3.3 Experimental results

**3.3.1 Measurement of $^{177}$Lu.**   Fig 9c shows the time lag distribution of the twofold coincidence events. In this study, the TDC Cut of $\pm400$ ns was applied, and off-timing events were excluded from the analysis as accidental coincidental events. Fig 9d presents the total energy spectrum of the selected events after the TDC Cut. From this spectrum, the total absorption peaks at 113 keV and 208 keV originating from $^{177}$Lu were confirmed. However, it is clear that both total absorption peaks contain background components and are different from the results of the simulation shown in Fig 3. In particular, at 113 keV, the signal-to-noise (S/N) ratio is further reduced by the Compton events at 208 keV. Therefore, to select events to be used in image reconstruction, the ADC Cut of $\pm2\sigma$ was applied as indicated by the blue and red shaded area in Fig 9d, i.e., $208\pm28$ keV and $113\pm20$ keV calculated from the energy resolution $\sigma(E) = 1.001E^{0.4963}$ obtained by actual measurement of the counter (3.5 cm $CaF_2$(Eu) + PMT), which we developed in this study.

First, we investigated which energy window selection would be the most effective for $^{177}$Lu imaging. Fig 10 shows the reconstructed images obtained using the image sharpening technique. From left to right, the images represent the results for 208 keV, 113 keV, and 113 + 208 keV. From top to bottom, the measurement times are 1 min, 10 min, and 60 min. Here, the peak intensity of the reconstructed image corresponds to the number of projected events. At 208 keV, we were able to obtain an image with a sharp gamma-ray peak and with the background sufficiently suppressed to a level of zero in about 10 min. Here, the angular resolution $\sigma$ and CNR were 12 degrees and 28, respectively, for the image with a measurement time of 60 min, and these values were equivalent to the simulation results. To further evaluate the time dependence of the reconstructed image, the angular resolution $\sigma$ and CNR at 208 keV were analyzed every 2 min over a measurement period of 60 min (Fig 11). The angular resolution $\sigma$ rapidly improved within the first few minutes and gradually approached the value obtained from the 60-min measurement. In contrast, the CNR increased with measurement time. However, at 113 keV, although a peak appeared at about 10 min, the contrast was lower than in the simulation results. This is due to a poor S/N ratio relative to the simulation, as shown in the energy spectrum in Fig 9d, where the energy region around the 113 keV peak is further contaminated due to the effect of overlapping Compton components from the 208 keV gamma rays emitted by $^{177}$Lu itself. Furthermore, at 113 + 208 keV, an improvement in image quality was expected due to the large amount of statistics, but the contrast was lower than that of the 208 keV gamma rays. These findings suggest that the use of 208 keV gamma-ray images is the most effective for $^{177}$Lu imaging.

We also performed image reconstruction using the conventional technique. The results are shown in Fig 12. We found that although the angular resolution was larger than that of the image sharpening technique and the background region could not be suppressed to a level of zero, it was possible to successfully identify the approximate direction of the radiation source in just under 1 min. Also, as is shown in Fig 10, the best results were obtained for the 208 keV gamma-ray image.

Furthermore, Fig 13 shows, as an example of a clinical site, a composite image of a 208 keV reconstructed image and an omnidirectional optical image taken with a Theta camera (Ricoh), where (a) and (b) represent the results obtained using the conventional technique and the image sharpening technique, respectively. The measurement times are 30 s for (a) and 10 min for (b). Here, the gamma-ray intensity above 70% is colored in red. The peak of the gamma-ray intensity coincided with the source direction (azimuth, elevation) = (0 degrees, 0 degrees), which confirmed the success of $^{177}$Lu visualization. For online measurements, both images can be displayed in real time: the presence and approximate direction of the $^{177}$Lu

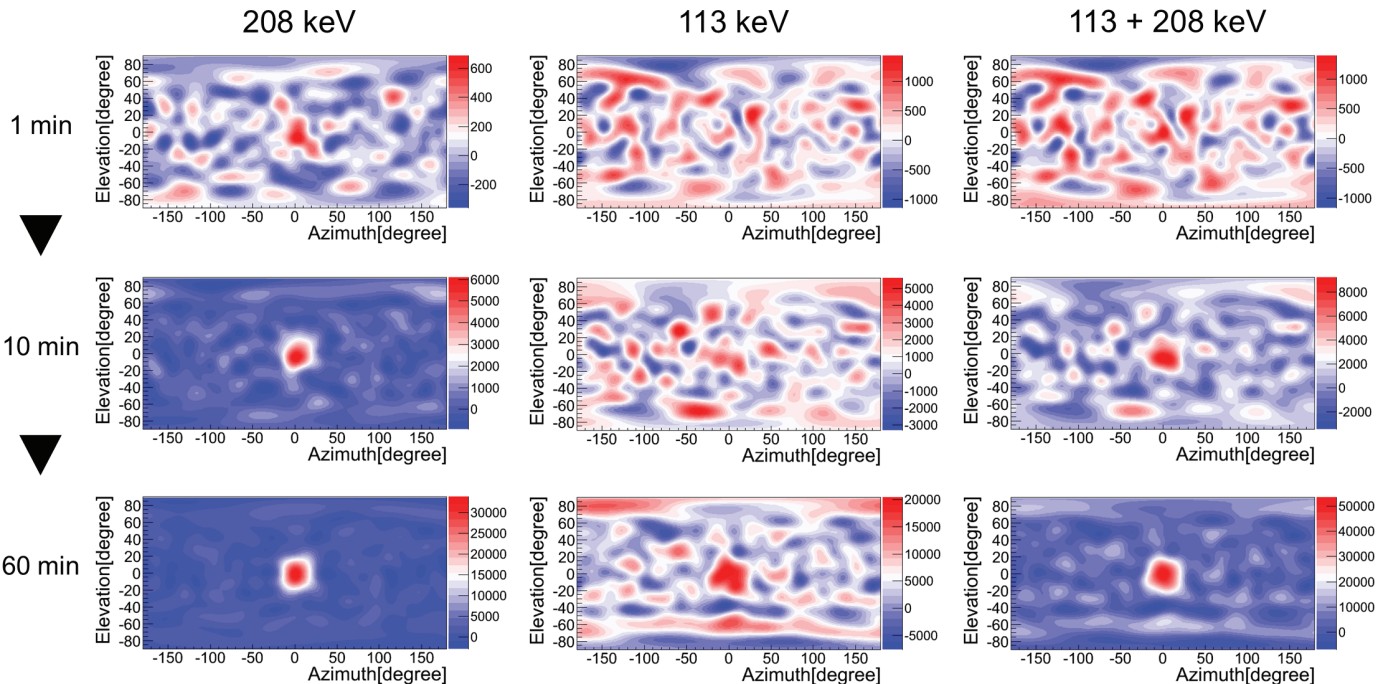

**Fig 10. Reconstructed images using the image sharpening technique.** The left, middle, and right columns represent the results for the energies of 208 keV, 113 keV, and 113 + 208 keV, respectively. The measurement times were 1 min (upper row), 10 min (middle row), and 60 min (lower rows), respectively.

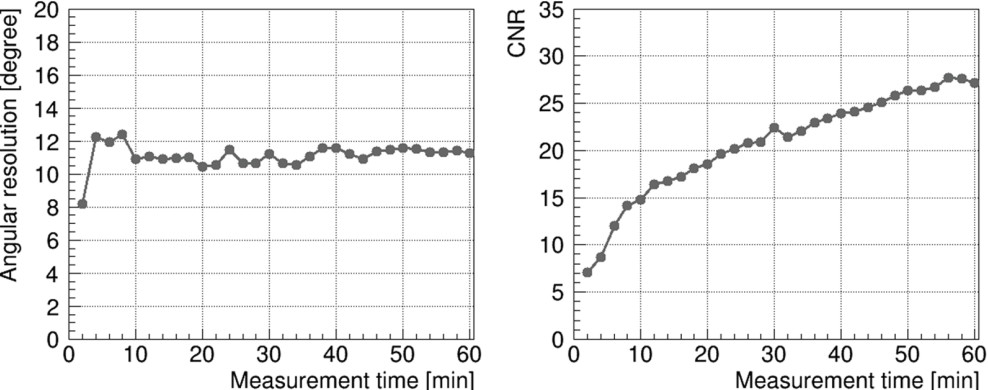

**Fig 11. Time dependence of angular resolution $\sigma$ and CNR in the reconstructed image at an energy of 208 keV.** Angular resolution $\sigma$ (left) and CNR (right) were evaluated every 2 min for 60 min using Eqs (4) and (5) to the results shown in Fig 10.

source can be immediately identified from the image using the conventional technique, and the image sharpening technique allows the display of a high-quality image over time. Here, the peak pixel value obtained from Fig 13b corresponds to the number of gamma rays measured, so it would possible to estimate the radioactivity by considering the absolute sensitivity shown in Fig 6 if a distance were given. To support this approach, we performed Monte Carlo simulations to evaluate the distance dependence of the absolute sensitivity. As a result, for source–detector distances $r \geq 30$ cm, the absolute sensitivity followed the inverse-square law

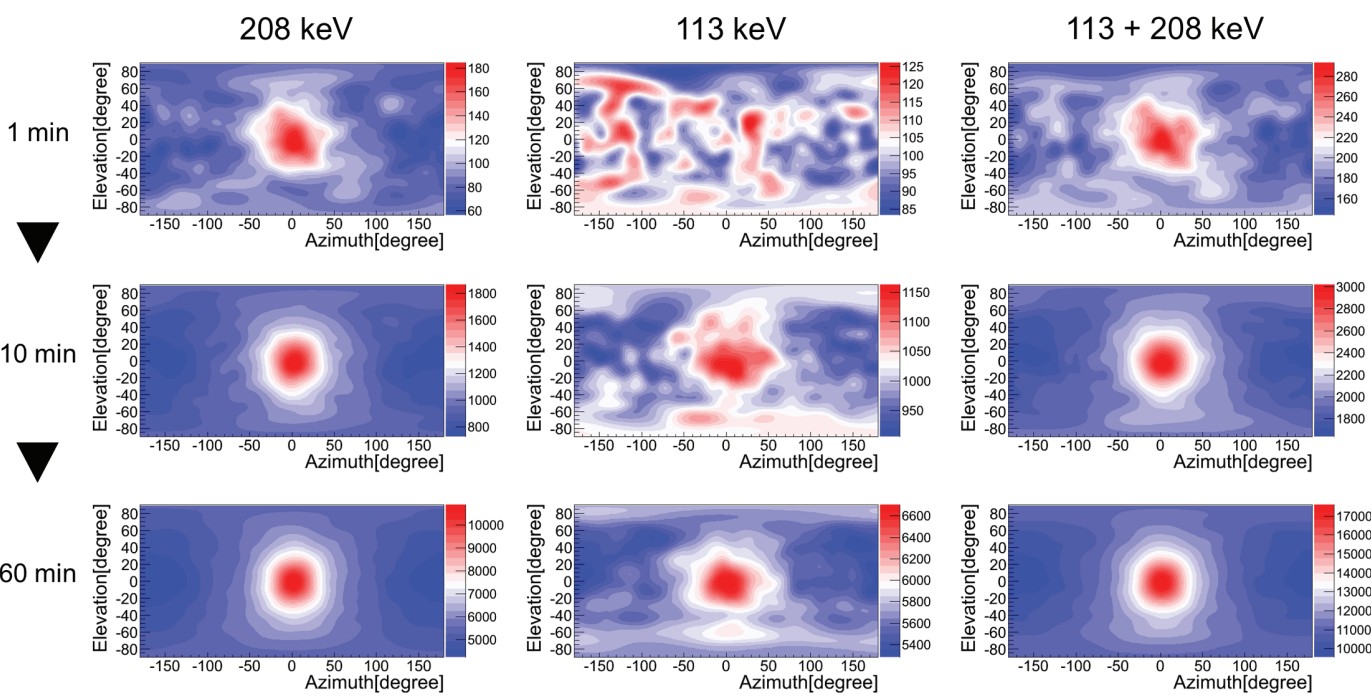

**Fig 12. Same as Fig 10, but with the use of the conventional technique.**

$(1/r^2)$, which was within statistical uncertainty, validating the applicability of distance-based correction in this range.

**3.3.2 Simultaneous measurement of $^{177}$Lu and $^{99m}$Tc.** Fig 14 shows the total energy spectrum of the twofold coincidence events obtained during the 30 min measurement. The spectrum shows the same total absorption peaks at 113 keV and 208 keV as were shown in the previous single-source results. However, the 141 keV gamma ray emitted from $^{99m}$Tc could not be separated due to overlap with the 113 keV peak. Here, the spectrum around 113 keV was fitted with a function as follows:

$$f(E) = p_0 \exp\left(-\frac{(E-113)^2}{2p_1{}^2}\right)$$
$$+ p_2 \exp\left(-\frac{(E-141)^2}{2p_3{}^2}\right) \tag{7}$$
$$+ \exp(p_4 E + p_5),$$

where $E$ represents gamma-ray energy in keV, and $p_0$ to $p_5$ are free parameters. The best-fit curves are also shown in Fig 14. This fitting successfully identified the total absorption peak at 141 keV. Based on these results, the energy window for 208 keV was set to $\pm 2\sigma$ (the red shaded area in Fig 14), whereas the energy window for the 141 keV peak was adjusted to $+2\sigma$ and $-\sigma$ (the green shaded area in Fig 14). This is because the 141 keV peak overlaps the 113 keV peak, thereby minimizing the effect of the 113 keV event.

Fig 15 shows a composite of the 141 keV and 208 keV reconstructed images and the omni-directional optical image, where (a) and (b) are superimposed with the conventional technique and the image sharpening technique, as in Fig 13. The measurement times are 2 min

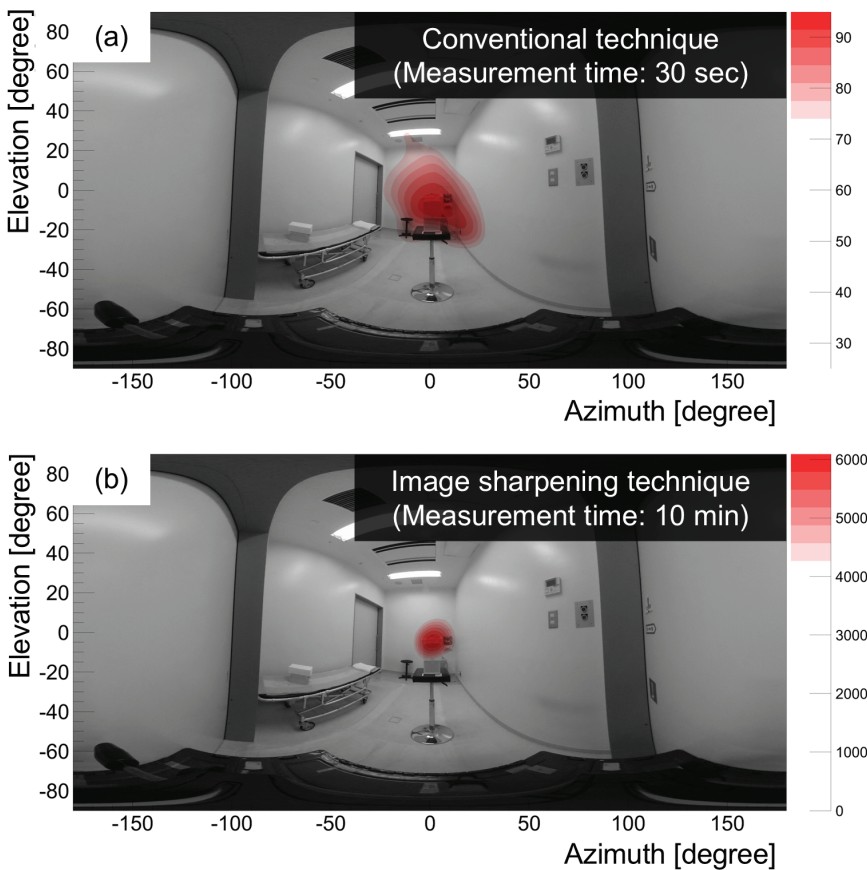

**Fig 13. 208 keV gamma-ray omnidirectional image superimposed on the optical image.** (a) The conventional technique (30 s data). (b) The image sharpening technique (10 min data). The red field indicates gamma-ray intensities having values greater than 70% of the peak value.

for (a) and 20 min for (b). Here, the gamma-ray intensity above 70% is shown in red color for $^{177}$Lu and in green for $^{99m}$Tc. We confirmed that the peaks of the gamma-ray intensity of both $^{177}$Lu and $^{99m}$Tc coincided with the direction of each radiation source. These results imply that when the 208 keV energy window was selected for $^{177}$Lu imaging, the presence of the $^{99m}$Tc source did not influence the $^{177}$Lu visualization process.

## 4 Discussion

In this study, we experimented with recently proposed rotating Compton camera technology to achieve the visualization of $^{177}$Lu-contaminated sites in nuclear medicine facilities. This technology has remarkable features in terms of cost-effectiveness, high sensitivity, and robustness. Our performance tests revealed that the use of 208 keV gamma rays is optimal for visualizing $^{177}$Lu with the developed detector. Furthermore, the detector was able to simultaneously visualize not only $^{177}$Lu-contaminated sites but also $^{99m}$Tc-contaminated sites, and these radionuclides are widely used in nuclear medicine facilities. This finding implies that this technology could realize environmental radiation monitoring with visualization functions in a wide range of nuclear medicine facilities.

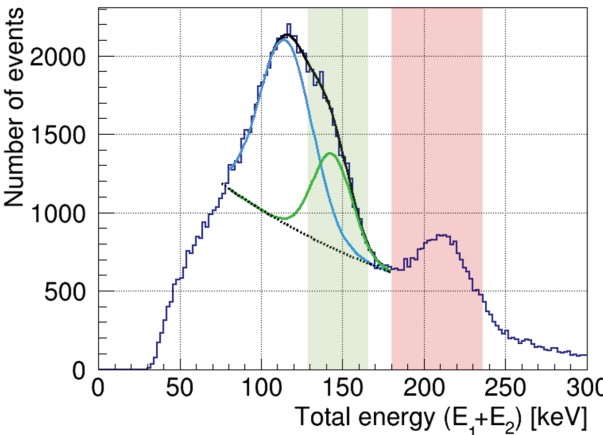

**Fig 14. Total energy spectrum for the simultaneous measurement of $^{177}$Lu and $^{99m}$Tc after the TDC Cut (the sum of $E_1 + E_2$).** The measurement time was 30 min. The black solid curve represents the best-fit curve obtained by fitting the data with a double Gaussian function having a peak energy of 113 keV (blue curve) and 141 keV (green curve) plus an exponential function (black dotted curve). The red shaded area represents the ADC Cut with a $\pm 2\sigma$ energy region for 208 keV, whereas the green shaded area represents the ADC Cut with a $+2\sigma$ and $-\sigma$ energy region for 141 keV to prevent the influence of events from the 113 keV peak due to its overlap with the 141 keV peak.

The proposed detector employs two types of image reconstruction techniques: the conventional technique and the image sharpening technique. The conventional technique enables rapid visualization, which makes it suitable for the immediate assessment of $^{177}$Lu-contaminated sites. Moreover, the image sharpening technique enhances the angular resolution, which is beneficial for precisely identifying the direction of radioactive contamination hotspots. Therefore, a two-step approach that uses both of these techniques would be preferable for application to actual $^{177}$Lu-contaminated sites. In the first step, the presence of radioactive contamination and the direction of the radiation source are quickly identified from the reconstructed image using the conventional technique, as well as from the trigger rate and energy spectrum. This information is important for making rapid initial decisions. Here, the key point is that this detector plays a crucial role in the initial phase by rapidly identifying the highest radiation hotspot and using this information to quickly initiate the first step of radioactive contamination removal. The highest radiation hotspot is identified first, and then decontamination is carried out with a survey meter. If we then measure again, the strongest hotspot has disappeared, and the next hotspot, if any, can be identified. This process, repeated iteratively, represents the fastest response for radioactive contamination removal with the use of this detector. In the second step, by continuing the measurements, the reconstructed image using the image sharpening technique is used to obtain a more detailed gamma-ray intensity distribution. This technique allows the precise identification of radioactive contamination and its spread by extracting minute information that cannot be captured by the conventional technique. A further advantage of this technique is that the peak values of the reconstructed image are scaled using $k$ in Eq (3), which corresponds directly to the number of events used in the image reconstruction. After the distance to the source is known, the radioactivity can immediately be estimated semiquantitatively. By combining the conventional technique with the image sharpening technique, this detector fulfills the role of both the rapid detection and the detailed analysis of radiation sources. In actual on-site contamination source detection, this two-step approach could work effectively to achieve efficient radiation management. This two-step imaging strategy can also be considered adequate from the

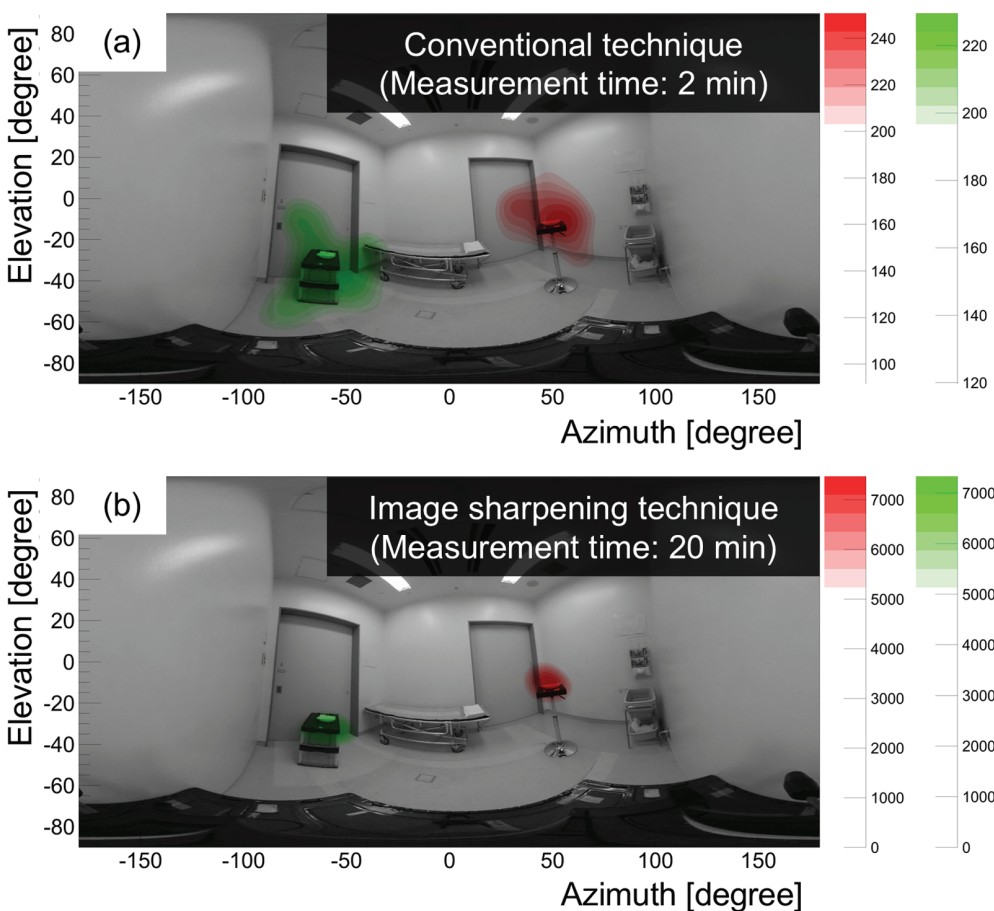

**Fig 15. 208 keV and 141 keV gamma-ray omnidirectional image superimposed on the optical image.** (a) The conventional technique (2 min data). (b) The image sharpening technique (20 min data). The red and green fields indicate gamma-ray intensities with values greater than 70% of the peak value, similar to that shown in Fig 13. Red corresponds to $^{177}$Lu (208 keV) and green to $^{99m}$Tc (141 keV).

perspective of the as low as reasonably achievable (ALARA) principles. The first step enables the rapid visualization of radioactive contamination, facilitating immediate decontamination while minimizing unnecessary radiation exposure to medical professionals and the public. The second step, which provides a high-resolution radioactivity distribution, identifies the direction of radioactive contamination in more detail. This minimizes the number and extent of decontamination operations required, thereby reducing the overall exposure and operational time. Accordingly, the proposed method offers a practical and realistic approach for radiation contamination management that aligns with the ALARA principles.

The detection efficiency of the developed rotating Compton camera for 208 keV gamma rays emitted from unsealed $^{177}$Lu-oxodotreotide in a vial was estimated to be 0.30 cps/MBq at 1 m from the actual measurements. However, the expected value obtained from considering the emission rate of 208 keV gamma rays from $^{177}$Lu ($\eta_\gamma = 0.11$) in the simulation results shown in Fig 6 was 0.50 cps/MBq at 1m, which is 40% greater than the measured value. This discrepancy is mainly due to self-absorption by the liquid containing $^{177}$Lu-oxodotreotide and the glass vial: the RI dose calibrator used in the performance test measures the radioactivity of the $^{177}$Lu that exists in the vial, but our detector measures the gamma rays that pass

through the vial without being self-absorbed after radioactive decay. However, because the composition and structure of the vial are not publicly disclosed, we performed Monte Carlo simulations to estimate the effect of self-absorption. In the simulation, the vial was assumed to be made of typical borosilicate glass ($SiO_2$: 80%, $B_2O_3$: 13%, $Na_2O$: 4%, $Al_2O_3$: 3%) and was modeled as a cylindrical shape with an outer diameter of 31.5 mm, an inner diameter of 28.5 mm, and a height of 43 mm. To more accurately replicate the measurement conditions, we assumed a volume source filled with 25 mL of water instead of a point source. The self-absorption effect due to the liquid and glass vials was approximately 20%, indicating that the observed discrepancy can be attributed to this factor. Nevertheless, other potential factors, such as differences in the actual vial composition, calibration errors in the RI dose calibrator, and variations in the source positioning during measurements, should also be considered. Another potential cause of this discrepancy is the reproducibility of the simulation. A key difference between the simulation described in Sect 2 and the detector setup described in Sect 3 is the presence of the PMT. Assuming that the PMT is made of aluminum (density: 2.7 $g/cm^3$), we conducted a Monte Carlo simulation with the PMTs arranged vertically, as shown in Fig 8b. The absolute sensitivity of the 208 keV gamma rays, normalized to the absolute sensitivity at 0 degrees elevation (Fig 7), decreased by approximately 4% and 20% at elevation angles of 0 degrees and 90 degrees, respectively. The influence of the PMT on the absolute sensitivity of the simulation also contributed to the discrepancy in the detection efficiency. Here, we note that for the measurement of the sealed $^{139}$Ce source emitting 166 keV gamma rays, where the self-absorption effects are negligible, the simulated and measured detection efficiencies are in good agreement within 10%. Moreover, during the actual measurements, the detection efficiency in the elevation direction is lower than that of the simulation results shown in Fig 7 when the elevation angle exceeds $\pm$60 degrees, and it is lower by 30% at 90 degrees and lower by 80% at –90 degrees. This is due to the shielding of gamma rays by the PMT and electronics, as shown in Fig 8. We remark that the solid angles at which the sensitivity decreases are sufficiently small (at 13.4%) compared with that in all directions. Furthermore, the absolute sensitivity is easily enhanced by reducing the interval between crystals: reducing the interval between the 3.5 cm cubic $CaF_2$ crystals from 10 to 7 cm slightly degrades the angular resolution $\sigma$ from 12 to 16 degrees for 208 keV gamma rays but increases the absolute sensitivity by a factor of two, which enables the realization of a more rapid response in identifying the contamination sites.

Although this study focused on demonstrating the feasibility of $^{177}$Lu imaging, the potential applications of this omnidirectional rotating Compton camera extend beyond the current scope. In particular, further developments could enhance its utility in a broad range of radiation environments. As is expected from Fig 6, combining CsI(Tl) and $CaF_2$(Eu) with the detector could potentially enable one detector to cover a wider energy range of 0.1–1.8 MeV. Furthermore, by considering the application of an iterative image reconstruction algorithm based on maximum likelihood expectation maximization [38] and 3D image reconstruction using measurement data from multiple directions, it may be possible to obtain more detailed information about contaminated areas. In addition to the low-dose imaging discussed in this study, the rotating Compton camera technology has great potential for high-dose imaging on the order of mSv/h in the actual decommissioning of the Fukushima Daiichi Nuclear Power Plant by employing small scintillator crystals. The details of the studies mentioned above will be presented in other papers in the near future.

## Supporting information

**S1 File. Source data, ROOT scripts, output images, and README file for Figs 5, 6, 7, and 11.**
(ZIP)

## Acknowledgements

This study was supported by the Open Source Consortium of Instrumentation (Open-It), Japan.

## Author contributions

**Conceptualization:** Hikari Tsukamoto, Hiroshi Muraishi, Ryoji Enomoto, Hideaki Katagiri, Mika Kagaya, Takara Watanabe, Takahiro Mizoguchi.

**Data curation:** Hikari Tsukamoto, Hiroshi Muraishi.

**Formal analysis:** Hikari Tsukamoto, Hiroshi Muraishi.

**Funding acquisition:** Hiroshi Muraishi, Takara Watanabe.

**Investigation:** Hikari Tsukamoto, Hiroshi Muraishi, Ryoji Enomoto, Hideaki Katagiri, Mika Kagaya, Takara Watanabe, Takahiro Mizoguchi, Masaya Fukumoto, Daisuke Kano.

**Methodology:** Hikari Tsukamoto, Hiroshi Muraishi, Ryoji Enomoto, Hideaki Katagiri, Mika Kagaya, Takara Watanabe, Takahiro Mizoguchi, Yusuke Watanabe, Kazuya Sakaguchi, Hiromichi Ishiyama.

**Project administration:** Hiroshi Muraishi, Ryoji Enomoto, Hideaki Katagiri.

**Resources:** Hikari Tsukamoto, Hiroshi Muraishi, Ryoji Enomoto, Hideaki Katagiri, Mika Kagaya, Takara Watanabe, Takahiro Mizoguchi, Daisuke Kano.

**Software:** Hikari Tsukamoto, Hiroshi Muraishi, Ryoji Enomoto, Hideaki Katagiri, Mika Kagaya, Takara Watanabe, Takahiro Mizoguchi.

**Supervision:** Hikari Tsukamoto, Hiroshi Muraishi, Ryoji Enomoto, Hideaki Katagiri, Mika Kagaya, Takara Watanabe, Takahiro Mizoguchi.

**Validation:** Hikari Tsukamoto, Hiroshi Muraishi.

**Writing – original draft:** Hikari Tsukamoto, Hiroshi Muraishi.

**Writing – review & editing:** Hikari Tsukamoto, Hiroshi Muraishi, Ryoji Enomoto, Hideaki Katagiri, Mika Kagaya, Takara Watanabe, Daisuke Kano, Yusuke Watanabe, Kazuya Sakaguchi.

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
