## [Decision Letter · Decision Letter 0]

3 Jun 2025

PONE-D-25-13108Development of an omnidirectional rotating Compton camera for imaging 177Lu radioactive contaminationPLOS ONE

Dear Dr. Tsukamoto,

Thank you for submitting your manuscript to PLOS ONE. After careful consideration, we feel that it has merit but does not fully meet PLOS ONE’s publication criteria as it currently stands. Therefore, we invite you to submit a revised version of the manuscript that addresses the points raised during the review process.

 Please submit your revised manuscript by May 16 2025 11:59PM. If you will need more time than this to complete your revisions, please reply to this message or contact the journal office at plosone@plos.org. When you're ready to submit your revision, log on to >https://www.editorialmanager.com/pone/ and select the 'Submissions Needing Revision' folder to locate your manuscript file.

We look forward to receiving your revised manuscript.

Kind regards,

Hesham M.H. Zakaly, Ph.D.

Academic Editor

PLOS ONE

Reviewers' comments:

Reviewer's Responses to Questions

**Comments to the Author**

1. Is the manuscript technically sound, and do the data support the conclusions?

Reviewer #1: Yes

Reviewer #2: Yes

Reviewer #3: Yes

Reviewer #4: Yes

2. Has the statistical analysis been performed appropriately and rigorously? 

Reviewer #1: I Don't Know

Reviewer #2: Yes

Reviewer #3: Yes

Reviewer #4: Yes

3. Have the authors made all data underlying the findings in their manuscript fully available?

Reviewer #1: Yes

Reviewer #2: Yes

Reviewer #3: Yes

Reviewer #4: No

4. Is the manuscript presented in an intelligible fashion and written in standard English?

Reviewer #1: Yes

Reviewer #2: Yes

Reviewer #3: Yes

Reviewer #4: Yes

5. Review Comments to the Author

Reviewer #1: Authors developed an omnidirectional rotating Compton camera that was capable of imaging low-level radioactive contamination caused by 177Lu-oxodotreotide. Novelty is proved by a attracting attention in nuclear medicine. By optimizing the crystal type and size,and optimizing the interval between crystals, the detector is able to adapt to a wide

range of environmental conditions, including observable gamma-ray energies, dose rates, and angular resolution. Monte Carlo simulations using Geant4 were conducted to optimize the configuration of the detector. Based on the results of the simulation, a prototype detector using six 3.5 cm cubic CaF2(Eu) crystals was developed for visualizing 177Lu-contaminated sites. The experimental results demonstrated that the detector could successfully visualize an unsealed 177Lu-oxodotreotide source with high sensitivity without being affected by gamma rays from 99mTc, which is also present in nuclear medicine facilities.

In my opinion text might be accepted for a publication in a Plos One journal.

Reviewer #2: The manuscript primarily discusses the research and development of an omnidirectional rotating Compton camera designed for imaging radioactive contamination from 177Lu. Monte Carlo simulations were conducted using Geant4 in a virtual setting to evaluate the detector's performance with various combinations of crystal types, sizes, and gamma-ray energies. The simulation findings indicated that a detector made of six 3.5-cubic-centimeter CaF2 crystals was the most suitable option. Following this, a prototype detector was created to visualize areas contaminated with 177Lu and 99mTc. Experimental results show that the rotating Compton camera technology excels in cost-effectiveness, sensitivity, and stability. The use of 208 keV gamma rays is ideal for imaging 177Lu, and this detector can simultaneously detect contamination from both 177Lu and 99mTc, which are frequently found in nuclear medicine facilities. This suggests its broad applicability in environmental radiation monitoring within such facilities. Its capability to quickly identify radioactive contamination could help minimize radiation exposure risks for medical staff and the public. However, prior to publication, the following issues need to be addressed:

In the course of developing the manuscript, what were the reasons for selecting the Monte Carlo simulation over alternative simulation techniques? Is the preference for this method attributable to its superior efficacy?

In the context of the Monte Carlo simulation, the selection of only three types of crystals—CsI, NaI, and CaF2—was made for specific experimental reasons, rather than an exhaustive evaluation of all available scintillator crystals. It is important to consider whether these three materials represent the optimal choices among scintillator crystals.

The manuscript indicates that the crystal size range utilized in the simulation spans from 0.5 to 4.0 cm. What rationale underpinned the selection of this specific range? What methodology was employed to ascertain this range? Furthermore, are there alternative size ranges that could potentially enhance the detector's performance?

The manuscript indicates that the detector is capable of concurrently visualizing the contamination of 177Lu and 99mTc. However, in practical applications, it is important to consider whether the simultaneous detection of these two contaminants may interfere with one another, potentially compromising the accuracy of the detection outcomes.

The manuscript discusses two methodologies for image reconstruction: the conventional technique and the image sharpening technique. In practical applications, which of these techniques is more prevalent? Which technique demonstrates greater utility? Additionally, have you contemplated the possibility of integrating these two approaches?

In the course of this research, what measures were implemented to ensure the accuracy and reliability of the experimental data? Were multiple repetitions of the experiments performed to mitigate potential errors? If so, what was the minimum error observed across these experiments?

The findings presented in the manuscript indicate that this detector has the potential to mitigate radiation exposure risks for both medical personnel and the general public. Consequently, what are the recommended protocols for the application of this detector in practical settings to effectively minimize these risks?

The manuscript does not provide an analysis of the detector's performance following extended usage. Could you clarify whether the detector's performance is expected to deteriorate over prolonged periods of use?

Reviewer #3: This manuscript presents the development of an omnidirectional rotating Compton camera designed to visualize low-level radioactive contamination caused by ^177^Lu-oxodotreotide in nuclear medicine facilities. The study combines Monte Carlo simulations (Geant4) with experimental validation to optimize detector configuration, focusing on crystal type, size, and rotational mechanics. The proposed detector demonstrates promising capabilities in selectively imaging ^177^Lu contamination while suppressing interference from 99m Tc, addressing a critical need for efficient radiation monitoring in clinical settings. The experimental methodology is comprehensive, covering simulations, prototype development, and performance tests under realistic conditions. However, while the study convincingly demonstrates feasibility, several aspects require further clarification or validation to strengthen scientific rigor and practical applicability.

(a) The Geant4 simulation employs the FTFP-BERT physics list. Why was this specific model chosen for low-energy gamma-ray interactions (e.g., 113–208 keV)? Are there validated benchmarks for

(b) ^177^Lu gamma-ray transport in CaF2 crystals?

(c) The energy resolution formula σ(E)=0.5746E^0.5736 is adopted from a prior CsI(Tl) study. How does this generalize to CaF2 (Eu), given differences in light yield and decay time (Table 2)? Provide experimental calibration data for the CaF2-PMT system.

(d) The discrepancy between simulated and measured detection efficiency (0.50 vs. 0.30 cps/MBq) is attributed to self-absorption in the vial. Include a Monte Carlo sub-study quantifying self-absorption effects for ^177^Lu in glass/liquid media to validate this claim.

(e) For simultaneous 177177Lu and 99m99mTc detection, the 113 keV and 141 keV peaks overlap. The 141 keV window is adjusted asymmetrically (-σ to +2σ). Provide a quantitative analysis of cross-contamination risks and how this adjustment minimizes false positives.

(f) The sensitivity drops sharply at ±60° elevation due to PMT shielding (Fig. 7). How does this angular dependency impact practical deployment in rooms with complex geometries (e.g., ceilings, floors)? Suggest design modifications to mitigate this limitation.

(g) The study mentions future work on maximum likelihood expectation maximization (MLEM). Why was filtered back-projection prioritized here? Include preliminary comparisons between the two methods to justify the current approach.

(h) Angular resolution and CNR values (e.g., σ = 12°, CNR = 28) are derived from 60-minute measurements. How do these metrics scale with shorter acquisition times (e.g., 1–10 minutes)? Provide error bars or confidence intervals for key parameters.

(i) Reducing crystal spacing from 10 cm to 7 cm reportedly doubles sensitivity but degrades angular resolution. Present a systematic trade-off analysis (sensitivity vs. resolution) to guide end-users in customizing the detector for specific applications.

(j) The compact metal-packaged PMTs introduce directional shielding. Quantify the gamma-ray attenuation caused by PMT materials (e.g., aluminum housing) using simulations or experimental measurements.

(k) The prototype weighs 7.2 kg (excluding battery). Discuss its portability in real-world scenarios (e.g., handheld operation, mounting on robotic platforms). Include data on vibration resistance or thermal stability during rotation.

(l) How does the detector’s performance compare to commercial survey meters or other Compton cameras (e.g., Polaris-H, Si/CdTe-based systems) in terms of sensitivity, angular resolution, and operational speed?

(m) The detector is proposed for high-dose environments (e.g., Fukushima). Has the radiation tolerance of CaF22(Eu) crystals and PMTs been tested under prolonged exposure (e.g., 1–10 mSv/h)?

(n) The study claims semiquantitative estimation using peak values and absolute sensitivity. Clarify the mathematical framework for converting reconstructed intensity to activity (Bq) and validate it with multi-distance measurements.

(o) The rotational motion "virtually" increases crystal count. Provide a mathematical proof or simulation demonstrating how rotation suppresses ghost artifacts across varying source geometries (e.g., multiple distributed sources).

(p) The two-step imaging strategy (conventional → sharpening) is proposed for contamination removal. How would this workflow integrate with existing radiation safety protocols (e.g., ALARA principles)? Include a case study or pilot deployment plan.

(q) The detector’s sensitivity drops above 300 keV for CaF22(Eu) (Fig. 6). For broader applicability, discuss hybrid crystal configurations (e.g., CaF22 + CsI) and provide preliminary simulation results.

(r) While ethical approval was deemed unnecessary, describe measures taken to ensure operator safety during experiments with unsealed 177177Lu sources (e.g., contamination control, dosimetry monitoring).

Reviewer #4: In this article, Tsukamoto et al. describes the development and performance of a Compton camera that is capable of detecting radiation from different directions (omnidirectional) to achieve position-sensitive detection (imaging) of the source of radiation. The Compton camera assembly consists of six scintillator crystal cubes, each mounted on a photomultiplier tube, with three facing upwards and three facing downward relative to an xy plane, to achieve bidirectional detection capabilities. The type and size of the crystal scintillators as well as the number and geometric arrangement of these detector elements in an array was optimized using Monte Carlo computer simulations. The authors also simulated the image reconstruction capabilities of this camera and the predicted sensitivity of each crystal type for different simulated gamma ray energies. When also considering cost and portability, the simulation results led to choosing europium-doped calcium fluoride crystal cubes of 3.5x3.5x3.5 cm, with six of these placed in an octahedral orientation. The authors next constructed a prototype based on the simulation results, mounting six 3.5 cm CaF2(Eu) crystals into the optimized octahedral array geometry. The entire assembly is mounted on a dual angle (azimuth and elevation) rotational stage, which achieves omnidirectionality and the ability for the Compton camera to image its surroundings. Additional post-acquisition data processing applying an image sharpening algorithm is used to reduce background noise and enhance the image quality in terms of sharpness, with a tradeoff in processing time.

This omnidirectional Compton camera prototype is demonstrated by imaging a lutetium-77 contaminated site and another site with both Lu-77 and Tc-99m, the latter being the most widely-used radioisotope currently used in medical facilities. The prototype was successfully able to locate the radioactive sources within a couple of minutes using conventional image reconstruction, and subsequently achieve higher spatial resolution in 10s of minutes once the post-processing sharpening algorithm was applied. Combined with the low cost and overall small dimensions of the device, this work effectively demonstrates the utility of the Compton camera for detecting the presence of exposed radioisotopes and pinpoint their location.

This work communicates in detail the evaluation and construction of the Compton camera, and demonstrates a working prototype that effectively images both one and two radiation sources as would be encountered in a “real world” scenario. The language is well-articulated and clear to follow the progression of thoughts from design to implementation. This paper is recommended for publication, after the authors consider some minor revisions, detailed below.

1. It is unclear if the designs for their Camera have been provided as supporting information, or the data used for their analysis been made openly available. These pieces of information are recommended prior to accepting this work for publication.

2. The europium-doped calcium fluoride crystals are sensitive to gamma radiation, however the authors also mention other radiopharmaceuticals (Table 1), two of which are not gamma emitters (yttrium-90 and strontium-89). How easy would it be to replace the scintillator crystal with one that would detect the beta radiation from these radioisotopes, and would the performance characteristics be expected to be significantly different for beta as is shown in their work for gamma? It would seem as if a scintillator that was sensitive to beta emissions would be capable of detecting/imaging more radioisotopes that are listed on Table 1 than the gamma-sensitive crystal used in this work. Please comment.

3. The authors note that the imaging performance of this camera is expected to be accurate to approximately 30 cm for a gamma source that is 1.5 meters away. Presumably, this accuracy could be improved by either moving the camera closer to the radiation source, or placing it in another location in the same room to “triangulate” the position of the radiation source. Can these points be addressed in the discussion?

4. The authors note that some amount of gamma radiation is obscured by the PMT assembly and associated electronics (line 438 and Fig 8) and suggest that the sensitivity of the detection array could be improved by bringing the scintillators closer together, at a cost of angular resolution. Could there be a way to reconfigure the placement of the scintillators “on the fly” using a mechanically-translatable assembly (such as a Hoberman-type octahedral scaffold), or would the additional mechanical braces needed for such an assembly compromise the sensitivity?

6. PLOS authors have the option to publish the peer review history of their article (what does this mean?). If published, this will include your full peer review and any attached files.

Reviewer #1: No

Reviewer #2: No

Reviewer #3: No

Reviewer #4: No

---

## [Author Response · Author response to Decision Letter 1]

Dear Prof. Hesham M.H. Zakaly

Thank you very much for your correspondence in regarding to the reviewers' comments. We have revised the manuscript based on these comments. Below, we have responded to each comment from the reviewers #2, #3, and #4. We hope that we have replied satisfactorily to all their questions.

We look forward to publishing our manuscript in PLOS ONE.

Sincerely yours,

Hikari Tsukamoto

Reply to reviewer #1:

N/A

Reply to reviewer #2:

Thank you very much for your comments. We have revised our manuscript accordingly. Revisions are indicated in blue font in the revised manuscript. We have also provided responses to your comments below (shown in blue fonts). Please note that some figure numbers have changed in the revised manuscript because of the added figure (Figure 11).

(a) In the course of developing the manuscript, what were the reasons for selecting the Monte Carlo simulation over alternative simulation techniques? Is the preference for this method attributable to its superior efficacy?

Analytical approaches or statistical models can be considered alternative methods; however, interactions between radiation and matter are highly complex. Therefore, we used Monte Carlo simulation in this study. This simulation can flexibly model intricate geometrical conditions, making it particularly suitable for analyzing various crystal configurations. Based on this advantage, Monte Carlo simulation is the most applicable and practical approach for this study.

(b) In the context of the Monte Carlo simulation, the selection of only three types of crystals—CsI, NaI, and CaF2—was made for specific experimental reasons, rather than an exhaustive evaluation of all available scintillator crystals. It is important to consider whether these three materials represent the optimal choices among scintillator crystals.

We believe that the three materials (CsI, NaI, and CaF₂) are appropriate for this study.

In the Compton camera developed in this study, the angular resolution σ is defined by the root sum square of the three components derived through error propagation as follows:

σ = √(σ_E² + σ_Geo² + σ_Dop²)

Here, σ_E is the energy resolution of the scintillator crystals, σ_Geo is the geometrical factor determined by the crystal size and arrangement, and σ_Dop is the Doppler broadening limit. Among these variables, σ_Geo is the dominant factor in the Compton camera system. Because σ_Geo primarily depends on the geometrical configuration rather than the scintillator material, the type of scintillator has minimal influence on the overall angular resolution.

Furthermore, sensitivity at different gamma-ray energies, as shown in Figure 5, is influenced by the effective atomic number (Z_eff) of the scintillator. Therefore, we consider it reasonable to use three widely available and cost-effective scintillator crystals with different Z_eff values to assess the Compton camera system performance rather than evaluating all possible scintillator materials comprehensively.

(c) The manuscript indicates that the crystal size range utilized in the simulation spans from 0.5 to 4.0 cm. What rationale underpinned the selection of this specific range? What methodology was employed to ascertain this range? Furthermore, are there alternative size ranges that could potentially enhance the detector's performance?

As shown in Figure 5, the angular resolution (σ) and contrast-to-noise ratio (CNR) are independent of the crystal size when the interval between crystals is scaled proportional to the crystal size. Meanwhile, the crystal size considerably affects the absolute sensitivity of the detector. Although the absolute sensitivity improves with increasing the crystal size, it plateaus as the crystal size exceeds 4.0 cm. Therefore, we only report the results for crystal sizes up to 4.0 cm.

Notably, doubling the crystal size necessitates a proportional doubling of the intercrystal spacing, which, in turn, requires the doubling of the overall detector size. Such an increase in crystal size is impractical from the design and implementation perspectives. In addition, the cost of scintillator crystals increases proportionally with their volume. Therefore, extending the crystal size beyond 4.0 cm is not realistic in terms of detector design and cost-effectiveness.

(d) The manuscript indicates that the detector is capable of concurrently visualizing the contamination of 177Lu and 99mTc. However, in practical applications, it is important to consider whether the simultaneous detection of these two contaminants may interfere with one another, potentially compromising the accuracy of the detection outcomes.

As described in the Discussion section (pages 13–14, lines 416–419), using a 208-keV gamma ray for 177Lu imaging ensures that there is no interference from gamma-ray events emitted from 99mTc. Because the primary focus of this study is 177Lu imaging, this concern lies outside the main scope of the paper. However, as correctly noted, if 99mTc imaging is considered, it may be affected by Compton scattering originating from 177Lu, as shown in the energy spectrum in Figure 14. The degree of this interference depends on the relative intensity of 177Lu compared to 99mTc.

(e) The manuscript discusses two methodologies for image reconstruction: the conventional technique and the image sharpening technique. In practical applications, which of these techniques is more prevalent? Which technique demonstrates greater utility? Additionally, have you contemplated the possibility of integrating these two approaches?

As mentioned in the second paragraph of the Discussion section (page 14, lines 422–451), the Compton camera developed in this study supports conventional and image-sharpening techniques, allowing users to select the most appropriate method depending on the application and radioactive contamination environment. This flexibility makes the system suitable for cases where either rapid decontamination or a high angular resolution is prioritized. Each technique has advantages and limitations and should be considered an independent methodology. Furthermore, as discussed in the fourth paragraph of the Discussion section (page 15, lines 509–513), we consider the maximum likelihood expectation maximization (MLEM) method as an extension or integration of these two approaches. However, another reviewer has also asked about methodology. Therefore, we have described the two-step imaging strategy employed in this study from the perspective of the as low as reasonably achievable (ALARA) principle in the Discussion section of the revised manuscript (page 14, lines 451–460) as follows.

“This two-step imaging strategy can also be considered adequate from the perspective of the as low as reasonably achievable (ALARA) principles. The first step enables the rapid visualization of radioactive contamination, facilitating immediate decontamination while minimizing unnecessary radiation exposure to medical professionals and the public. The second step, which provides a high-resolution radioactivity distribution, identifies the direction of radioactive contamination in more detail. This minimizes the number and extent of decontamination operations required, thereby reducing the overall exposure and operational time. Accordingly, the proposed method offers a practical and realistic approach for radiation contamination management that aligns with the ALARA principles.”

(f) In the course of this research, what measures were implemented to ensure the accuracy and reliability of the experimental data? Were multiple repetitions of the experiments performed to mitigate potential errors? If so, what was the minimum error observed across these experiments?

Thank you for your comment. The energy resolution of the scintillator crystals used in this study is approximately 18.6% FWHM at 166 keV. Error arising due to the temperature dependence of the PMT, which is in the order of few percents, is sufficiently small relative to this energy resolution value and does not significantly affect the experimental results as a systematic error. Furthermore, before measurements, each counter is calibrated using a known energy source. For more details, please refer to Refs. [26] and [31]. In addition, to minimize the impact of fluctuations due to electronics or power instability, the power supply and preamplifier board are custom-developed. Multiple measurements are conducted in the laboratory using a 139Ce (166 keV, 174 kBq) source, and the results are reproducible within the statistical error range.

(g) The findings presented in the manuscript indicate that this detector has the potential to mitigate radiation exposure risks for both medical personnel and the general public. Consequently, what are the recommended protocols for the application of this detector in practical settings to effectively minimize these risks?

We have described an example of a protocol in the Discussion section (page 14, lines 422–460) as follows:

“The proposed detector employs two types of image reconstruction techniques: conventional and image sharpening techniques. The conventional technique enables rapid visualization, making it suitable for the immediate assessment of 177Lu-contaminated sites. Moreover, the image sharpening technique enhances the angular resolution, which is beneficial for precisely identifying the direction of radioactive contamination hotspots. Therefore, a two-step approach that uses both of these techniques would be preferable for application to actual 177Lu-contaminated sites...” However, practical applications, such as measurements using the detector in RI rooms or “special measures patient room,” have not yet been conducted. To further explore the development of appropriate protocols, an ethical review will be involved, and we plan to take the appropriate steps moving forward.

(h) The manuscript does not provide an analysis of the detector's performance following extended usage. Could you clarify whether the detector's performance is expected to deteriorate over prolonged periods of use?

In general, conventional detectors rely on calibration constants for energy determination, and these constants may change over extended periods of use. However, the developed Compton camera is designed to minimize such effects by allowing its calibration before each measurement. This is achieved by switching to a single-trigger mode, which enables entry into a dedicated calibration mode. Thus, performance fluctuations due to long-term usage are effectively suppressed.

As rightly pointed out, if the detector is used in high-dose-rate environments, such as medical accelerator facilities, where gamma-ray energies exceed 1 MeV, the scintillator can become radioactivated because of the presence of neutrons, increasing background and potential performance degradation. However, this study focuses on applications under nuclear medicine settings, which do not involve such high-energy or high-dose conditions. Therefore, performance degradation due to prolonged use is considered negligible under the usage conditions assumed in this study.

Furthermore, we have modified some parts by following the suggestions from another reviewer as described below.

(1) On page 9, lines 265–267, we added a statement to the part of “4. Discussion”:

“Notably, the simulation results shown above were consistent even when detailed simulations were performed using a physics list, such as LIVEMORE [35], which was specifically designed for low-energy gamma rays.”

(2) On page 9, lines 267–274, we added a statement to the part of “2.4. Simulation results”:

“Furthermore, as described in Section 2.1, the energy resolution σ(E) obtained by measuring CsI(Tl) was employed in simulations using NaI(Tl) and CaF₂(Eu). Nonetheless, the simulation results were consistent even when the energy resolution obtained by measuring 3.5 cm cubic CaF₂(Eu) coupled with a metal-packaged PMT (σ(E)=1.001E^0.4963, which is 10% larger than that for CsI(Tl)), was used. This implies that the angular resolution of the Compton camera used in this simulation is dominated by the geometrical factor, consistent with a previous report [32].”

(3) On pages 14–15, lines 470–481, we added a statement to the part of “4. Discussion”:

“However, because the composition and structure of the vial are not publicly disclosed, we performed Monte Carlo simulations to estimate the effect of self-absorption. In the simulation, the vial was assumed to be made of typical borosilicate glass (SiO_2:80%, B_2 O_3:13%, Na_2 O:4%, Al_2 O_3:3%) and was modeled as a cylindrical shape with an outer diameter of 31.5 mm, an inner diameter of 28.5 mm, and a height of 43 mm. To more accurately replicate the measurement conditions, we assumed a volume source filled with 25 mL of water instead of a point source. The self-absorption effect due to the liquid and glass vials was approximately 20%, indicating that the observed discrepancy can be attributed to this factor. Nevertheless, other potential factors, such as differences in the actual vial composition, calibration errors in the RI dose calibrator, and variations in the source positioning during measurements, should also be considered.”

(4) On page 11, lines 351–355, we added a statement to the part of “3.3.1. Measurement of 177Lu”:

“To further evaluate the time dependence of the reconstructed image, the angular resolution σ and CNR at 208 keV were analyzed every 2 min over a measurement period of 60 min (Fig. 11). The angular resolution σ rapidly improved within the first few minutes and gradually approached the value obtained from the 60-min measurement. In contrast, the CNR increased with measurement time.”

(5) On page 15, lines 481–490, we added a statement to the part of “4. Discussion”:

“Another potential cause of this discrepancy is the reproducibility of the simulation. A key difference between the simulation described in Section 2 and the detector setup described in Section 3 is the presence of the PMT. Assuming that the PMT is made of aluminum (density: 2.7 g/cm3), we conducted a Monte Carlo simulation with the PMTs arranged vertically, as shown in Fig 8(b). The absolute sensitivity of the 208 keV gamma rays, normalized to the absolute sensitivity at 0 degrees elevation (Fig 7), decreased by approximately 4% and 20% at elevation angles of 0 degrees and 90 degrees, respectively. The influence of the PMT on the absolute sensitivity of the simulation also contributed to the discrepancy in the detection efficiency.”

(6) On page 3, lines 73–80, we added a statement to the part of “1. Introduction”:

“As an example of a gamma-ray detector for environmental radiation imaging, Polaris-H was developed by the University of Michigan and commercialized by H3D [23]. It uses CdZnTe (CZT) and offers excellent energy resolution (ΔE/662 keV= 1.1% FWHM). This system can achieve relatively high sensitivity as a Compton camera at energies above 250 keV, making it particularly useful for identifying radioactive contamination in nuclear power plants. However, at energies below 250 keV, it operates as a coded-aperture gamma-ray camera, which limits its field of view and sensitivity.”

(7) On page 12, lines 383–387, we added a statement to the part of “3.3.1. Measurement of 177Lu”:

“To support this approach, we performed Monte Carlo simulations to evaluate the distance dependence of the absolute sensitivity. As a result, for source–detector distances 𝑟 ≥ 30 cm, the absolute sensitivity followed the inverse-square law (1/r2), which was within statistical uncertainty, validating the applicability of distance-based correction in this range.”

(8) On page 10, lines 322–326, we added a statement to the part of “3.2. Performance tests”:

“To prevent radioactive contamination in case of vial or syringe damage, the 177Lu and 99mTc sources were each placed on trays lined with a filter pa

---

## [Decision Letter · Decision Letter 1]

16 May 2025

Development of an omnidirectional rotating Compton camera for imaging 177Lu radioactive contamination

PONE-D-25-13108R1

Dear Dr. Tsukamoto,

We’re pleased to inform you that your manuscript has been judged scientifically suitable for publication and will be formally accepted for publication once it meets all outstanding technical requirements.

Kind regards,

Hesham M.H. Zakaly, Ph.D.

Academic Editor

PLOS ONE

Additional Editor Comments (optional):

Reviewers' comments:

Reviewer's Responses to Questions

**Comments to the Author**

1. If the authors have adequately addressed your comments raised in a previous round of review and you feel that this manuscript is now acceptable for publication, you may indicate that here to bypass the “Comments to the Author” section, enter your conflict of interest statement in the “Confidential to Editor” section, and submit your "Accept" recommendation.

Reviewer #2: All comments have been addressed

Reviewer #4: All comments have been addressed

2. Is the manuscript technically sound, and do the data support the conclusions?

Reviewer #2: Yes

Reviewer #4: Yes

3. Has the statistical analysis been performed appropriately and rigorously? 

Reviewer #2: Yes

Reviewer #4: Yes

4. Have the authors made all data underlying the findings in their manuscript fully available?

Reviewer #2: Yes

Reviewer #4: Yes

5. Is the manuscript presented in an intelligible fashion and written in standard English?

Reviewer #2: Yes

Reviewer #4: Yes

6. Review Comments to the Author

Reviewer #2: The authors have thoroughly and successfully addressed the concerns raised in the manuscript, which now largely fulfills the criteria for publication in PLoS One. Below are the specific points that the reviewer feels require further clarification (only the aspects that the reviewer finds unclear):

This manuscript presents a formula for energy resolution utilized in the simulation process (e.g., "σ" ("E" )="0.5746E0.5736" ), derived from particular experiments. Therefore, can this formula still be used under varying experimental conditions or with different equipment? What is its range of applicability? Is there a need for recalibration or adjustment?

This manuscript discusses various factors, including self-absorption and the effects of the PMT, when examining the differences in detection efficiency. Do these factors interact with one another, resulting in a more intricate effect on detection efficiency?

This manuscript states that the detector's performance has been assessed via simulations and experiments. Nevertheless, in real-world applications, could there be variations in the performance of detectors made in different production runs? What measures can be taken to guarantee that the performance of mass-produced detectors remains consistent?

Reviewer #4: Thank you for addressing all of the concerns. The peer-review was quite intensive and the authors did an excellent job of respectfully responding to as many issues that would directly improve upon their work.

7. PLOS authors have the option to publish the peer review history of their article (what does this mean?). If published, this will include your full peer review and any attached files.

Reviewer #2: No

Reviewer #4: No

---

## [Editor Report · Acceptance letter]

PONE-D-25-13108R1

PLOS ONE

Dear Dr. Tsukamoto,

I'm pleased to inform you that your manuscript has been deemed suitable for publication in PLOS ONE. Congratulations! Your manuscript is now being handed over to our production team.

Kind regards,

on behalf of

Dr. Hesham M.H. Zakaly

Academic Editor

PLOS ONE